**On-flight intercomparison of three miniature aerosol absorption sensors using Unmanned Aerial Systems (UAS)**

Michael Pikridas[1], Spiros Bezantakos[1], Griša Močnik[2,3], Christos Keleshis[1], Fred Brechtel[4], Iasonas Stavroulas[1,5], Gregoris Demetriades[1], Panayiota Antoniou[1], Panagiotis Vouterakos[1], Marios Argyrides[1], Eleni Liakakou[5], Luka Drinovec[2,3], Eleni Marinou[6,7], Vassilis Amiridis[6], Mihalis Vrekoussis[1,8,9], Nikolaos Mihalopoulos[1,5] and Jean Sciare[1]

[1]*Energy Environment and Water Research Center, The Cyprus Institute, Nicosia 1645, Cyprus*

[2]*Aerosol d.o.o., 1000 Ljubljana, Slovenia*

[3]*Jozef Stefan Institute, 1000 Ljubljana, Slovenia*

[4]*Brechtel Manufacturing Inc., 1789 Addison Way, Hayward, CA 94544 U.S.A.*

[5]*Institute for Environmental Research and Sustainable Development, National Observatory of Athens, 15236, Athens, Greece*

[6]*Institute for Astronomy, Astrophysics, Space Applications and Remote Sensing (IAASARS), National Observatory of Athens (NOA), Athens, Greece*

[7]*Institute of Atmospheric Physics, German Aerospace Center (DLR), 82234 Weßling, Oberpfaffenhofen, Germany*

[8]*Institute of Environmental Physics, University of Bremen, Otto-Hahn-Allee 1, D-28359 Bremen, Germany*

[9]*Center of Marine Environmental Sciences - MARUM, D-28359 Bremen, Germany*

The present study investigates and compares the ground and in-flight performance of three miniaturized aerosol absorption sensors integrated on-board of small size Unmanned Aerial Systems (UAS). These sensors were evaluated during two contrasted field campaigns performed at an urban site, impacted mainly by local traffic and domestic wood burning sources (Athens, Greece), and at a remote regional background site, impacted by long-range transported sources including dust (Cyprus Atmospheric Observatory, Agia Marina Xyliatou, Cyprus).

The miniaturized sensors were first intercompared at the ground-level against two commercially available instruments used as a reference. The measured signal of the miniaturized sensors was converted into the absorption coefficient and equivalent black carbon concentration (*eBC*). When applicable, signal saturation corrections were applied, following the suggestions of the manufacturers. The aerosol absorption sensors exhibited similar behavior against the reference instruments during the two campaigns, despite the diversity of the aerosol origin, chemical composition, sources, and concentration levels. The deviation from the reference, during both campaigns, concerning (*eBC*) mass was less than 8% while for the absorption coefficient was at least 15%. This indicates that those sensors that report black carbon mass are tuned/corrected to measure more accurately *eBC* rather than the absorption coefficient.

The overall potential use of miniature aerosol absorption sensors on-board small UAS is also illustrated. UAS-based absorption measurements were used to investigate the vertical distribution of *eBC* over Athens up to 1 km above sea level during January 2016, exceeding the top of the planetary boundary layer (PBL). Our results reveal a heterogeneous boundary layer concentration of absorbing aerosol within the PBL intensified in the early morning hours due to the concurrent peak traffic emissions at

ground-level and the fast development of the boundary layer. After the full development of the PBL, homogenous concentrations are observed from the 100m a.g.l. to the PBL top.


## 1. Introduction

Atmospheric aerosol particles scatter and absorb solar radiation, thus directly affecting the radiative balance of the atmosphere (Haywood and Boucher, 2000). Their contribution to climate change is still associated with large uncertainties when
estimating their radiative forcing (RF) (Bond et al., 2013; IPCC, 2013). A major contributor to these uncertainties is the RF induced by black carbon (BC), which exhibits a relative standard deviation exceeding 40% among different numerical climate models (Myhre et al., 2013). The BC direct RF has been estimated to be 0.71 $Wm^{-2}$ with an uncertainty range of 0.08 to 1.27 $Wm^{-2}$ (Bond et al., 2013), while in a more
recent study it ranged from 0.14 to 1.19 $Wm^{-2}$ (90% confidence interval) with an average value of 0.53 $Wm^{-2}$ (Wang et al., 2016). Major factors responsible for the wide range of the BC's RF estimates include the inaccurately predicted BC emission rates, poorly understood interactions of BC with clouds, and the inaccuracy in representing its vertical distribution (Bond et al., 2013). In addition, BC has been identified to reduce
the albedo of snow surfaces (Hadley and Kirchstetter, 2012) and to suppress the turbulence of the boundary layer (Wilcox et al., 2016).

An array of techniques and instruments are employed worldwide with the aim to increase the spatial and temporal resolution of BC observations. The instrumentation employed is based on different operating principles, including off-line or near-real-time
methods for measuring Elemental Carbon (EC), such as thermal-optical reflectance/transmittance (cf. Lack et al., 2014 and references therein for more details) as well as on-line, real-time methods. The latter are mainly based on the aerosol light-absorbing properties of BC (cf. Moosmüller et al., 2009; Petzold et al., 2013; Lack et al., 2014 and references therein for more details).

Most of the aerosol absorption observations available in the literature are conducted at ground level. Consequently, they lack critical information regarding the vertical distribution of aerosol absorption a key parameter to constrain atmospheric models and accurately assess aerosol radiative forcing effects (Samset et al., 2018). One way to fill this gap is by conducting manned airborne aerial absorption measurements (Seinfeld et
al., 2004; Subramanian et al., 2010; Freney et al., 2014; Kassianov et al., 2018; Katich et al., 2018; Sedlacek et al., 2018). However, these are costly and cover a limited period of observations. In the pioneering work of Corrigan et al. (2008), vertical absorption profiles over the Indian Ocean were measured using parts from a standard (rack) size instrument onboard a medium scale (25-150 kg) unmanned aerial system (UAS). Since
then, the size and weight of absorption monitors have been reduced, and the use of lightweight miniaturized sensors on-board of small UAS or tethered balloons provides cost-effective alternatives able to fill the measurement gap and to enhance the vertical and temporal density of aerosol absorption observations. A UAS is defined as small if its gross weight is less than 25 kg (US Federal Aviation Administration, CFR 14).
Vertical aerosol absorption observations using small UAS or tethered balloons have been already conducted in different regions such as the Indian Ocean (Höpner et al., 2016), India (Bisht et al., 2016), the Arctic (Bates et al., 2013; Ferrero et al., 2016),

Italy (Ferrero et al., 2011; 2014), Poland (Chilinski et al., 2016), and China (Ran et al., 2016). These measurements can be further used to obtain the vertically resolved heating rate, including contributions from different sources and carbonaceous aerosol fractions (Ferrero et al., 2014; 2018). The employment of UAS in some of the above-mentioned campaigns proves to be a viable option to obtain information on aerosol absorption vertical distribution. Even though small UAS are subject to significant payload restrictions, compared to manned aircrafts, they have a distinct advantage over their manned counterparts in terms of relatively low platform and operation cost, capability to perform autonomous flight operations, the ability to fly closer to the ground with greater spatial accuracy and to collect spatially dense data (due to low speed operation) under reduced workload (Villa et al., 2016). In addition, they have the advantage of better controllability in comparison to balloons and zeppelins, since the latter are more delicate at stronger winds (Inoue et al., 2000; Jensen et al., 2007). In terms of instrumentation, ground-based aerosol absorption instruments have been qualified in many intercomparison studies (e.g. Müller et al., 2011). On the contrary, their miniaturized counterparts' behavior is yet poorly demonstrated in-flight. The measurement quality delivered by these sensors during flight is challenged by fast changes in pressure, temperature, and humidity, which are difficult to assess from concurrent ground-level measurements.

In this work, we focus on vertical distributions of aerosol absorption, measured with miniature absorption sensors onboard small and medium-size UAS during two intensive field campaigns at contrasted locations in the Eastern Mediterranean; an urban site (Athens, Greece) and a remote regional background site (Cyprus Atmospheric Observatory, CAO, Cyprus). The vertical distribution of aerosols in the Eastern Mediterranean is of particular importance because it lies at the crossroads of diverse air masses (Lelieveld et al., 2002) carrying aerosol of different composition, including mineral dust from Africa and the Middle East, pollution from Europe and the nearby Middle East, and marine aerosol (Erel et al., 2006; Gerasopoulos et al., 2007; Kalivitis et al., 2007). In addition, aerosol absorption measurements, either ground or aerial-based, are rather scarce in the region. The sites were selected to represent two different and contrasted sources of ambient aerosol, with high concentration levels of freshly emitted BC from traffic and/or biomass burning (domestic heating) in Athens and low concentrations of aged regionally transported aerosol, occasionally mixed with moderate levels of dust in Cyprus.

Aerosol vertical profiles were monitored using several types of fixed and rotary wing UAS. In addition to the aerial observations, three miniature attenuation monitors were also characterized against ground-based commercial instruments. Secondly, these miniature sensors were compared and characterized in-flight with different UASs and diverse absorbing aerosol concentrations and types.

## 2. Instrumentation
### 2.1 Unmanned Aerial System Types

Three types of UAS have been used in this study; they differ with respect to the payload, autonomy, wing type, and landing requirements. Their specifications and capabilities, described below, are summarized in Table 1. In addition and as mentioned before, UAS are characterized as small when their gross weight is less than 25 kg and

medium if their gross weight ranges between 25 - 150 kg (US Federal Aviation Administration, CFR 14). Despite having the ability to reach altitudes higher than 2 km above ground level (a.g.l.), the UASs were limited to 1 and 2 km during the Athens and Cyprus campaigns, respectively, due to restrictions posed by the civil aviation authorities.

### 2.1.1 UAS "Cruiser"

The Cruiser is a medium-size fixed-wing UAS (Table 1) with a payload capacity up to 12 kg which includes also the weight of the fuel to power the engine and the battery used for the instrumentation. The Cruiser's payload bay, available inside the UAS, measures 1.3 m $\times$ 0.23 m $\times$ 0.34 m (LxWxH). The UAS features a wingspan of 3.8 m. It has been configured with an internal combustion two-stroke engine placed in a push configuration enabling an altitude ceiling of 4 km and maximum take-off weight of 35 kg. Depending on payload and environmental conditions the Cruiser can reach a flying endurance up to 8 hours. During the flight, atmospheric sampling occurs at a velocity of 28±5 m s$^{-1}$ which is the typical cruising air-speed of this type of UAS. Under its current configuration the environmental conditions to ensure safe operation are limited to winds up to 13 m s$^{-1}$ and temperatures above the dew point in order to prevent icing on the engine's carburetor. The Cruiser is equipped with an autopilot system (Micropilot MP2128G2) which includes all the sensors and telecommunication systems (e.g. GPS, barometric altimeter, accelerometer, air-speed sensor, electronic compass, modems, antennas) that allows autonomous flights with real-time monitoring and control from the ground providing that predetermined flight plans are set. At any time, the UAS operator is able to modify the active flight plan in real-time. In addition, the system is capable of detecting faults and alter its flight plan accordingly (e.g. automatically return to home upon communication loss). The modular design of the Cruiser facilitates switching instruments between scientific missions provided that the total mass does not exceed the payload limit. To support its multi-instrument capability, a central data acquisition system built around the National Instruments controller, myRIO with a variety of interface possibilities and a Graphical User Interface (GUI) has been developed. The graphical programming language Labview (from National Instruments) has been utilized to develop the GUI with capabilities of real-time visualization of the instrumentation data as well as controlling and automation of the on-board instruments. All the instruments and avionics sensitive to vibration have been mounted into the Cruiser fuselage using special anti-vibration dampers in order to insulate them from the high-frequency oscillations produced by the UAS engine. Vibration insulation is essential in order to improve the flying reliability of the UAS as well as to keep the quality of the scientific measurements to its higher standards.

Due to the Cruiser's size, a flat (ideally paved) runway is required for take-off and landing. During the Cyprus campaign, the Cruiser was taking-off and landing on Cyprus Institute's private runway (see Fig. 1).

### 2.1.2 UAS "Skywalker X8"

The Skywalker X8 is a small delta-wing type UAS with an electric motor providing the propulsion. Made from foam, it is a much smaller and lower cost UAS compared to the Cruiser. Its wingspan is 2.10 m and its maximum take-off weight is about 5.5 kg. It can fly for approximately one hour up to 3 km altitude with a payload of c.a. 3 kg,

which includes the battery (14.8V Lithium Polymer, 9Ah) that powers the motor. This UAS is equipped with the same avionics as the Cruiser, maintaining all of its advanced automation characteristics. The Skywalker X8 can take-off using a bungee launcher catapult system and can land on its belly on any flat surface, thus minimizing the requirements for a specialized aerodrome.

### 2.1.3 UAS "Multicopter S1000+"

A modified version of the commercially available octocopter DJi S1000+ was used during the Athens campaign to overcome strong constraints related to a limited ground area for take-off and landing, and flying in the limited air space. This platform has been optimized to reach an altitude up to 1 km above sea level (a.s.l.) for a maximum take-off weight of 11 kg and a payload of 4 kg including the motor battery (22V Lithium Polymer, 22Ah). In order to ensure that sampling was not influenced by the turbulence created by the octocopter's blades, the sampling inlet was extended by 1 m out of the propeller downdraft. This distance ensured representative sampling while ascending. However, during descent, this length was not sufficient to avoid the created vortex when a columnar path was followed. During the Athens campaign, the landing site was near the edge of a cliff and inside an archaeological area where pedestrians could freely access (Fig. 2), prohibiting deviation from a columnar flight path. As a result, the quality of the descent flights was compromised at the expense of safety and thus only ascending flights are used in this work.

## 2.2 Aerosol absorption instrumentation

### 2.2.1 Principle of operation

The most widely used instruments for the determination of the aerosol absorption coefficient are filter photometers. They sample ambient air through a filter, where the sample is collected. The filter is illuminated and the light transmitted through the filter is measured. Transmission of the sample-laden filter is normalized to the transmission of the sample-free filter (reference signal) and the attenuation is calculated based on Eq. 1.

$$ATN(\lambda) = 100 \times \ln\left(\frac{I_{ref}(\lambda)}{I_{sample}(\lambda)}\right) \qquad Eq.\,1$$

where $I_{ref}(\lambda)$ and $I_{sample}(\lambda)$ are the reference and sample light intensities at the detectors under the filter, respectively, and $ATN(\lambda)$ the attenuation at wavelength $\lambda$. The attenuation rate $dATN(\lambda)/dt$ (calculated from consecutive measurements) determines the attenuation coefficient ($b_{atn}(\lambda)$) based on Eq. 2.

$$b_{atn}(\lambda) = \frac{A}{100Q}\frac{dATN(\lambda)}{dt} \qquad \textbf{Eq. 2}$$

where $A$ is the sample spot area, $Q$ the airflow rate and $dt$ the time period for which the attenuation change is considered. It typically equals to 1s for all the miniaturized instruments examined in this study. The instrument specific $b_{atn}(\lambda)$ can be converted to absorption coefficient $b_{abs}(\lambda)$, when accounting for the multiple scattering effects caused by the filter and/or by the sampled particles, together with the filter loading

effects that the latter are causing. Due to a lack of a reference method for providing the aerosol absorption coefficient and because every manufacturer is using different filter materials, several empirical corrections have been proposed in the literature (e.g. Weingartner et al., 2003; Virkkula et al., 2005; Collaud Coen et al., 2010; Ogren, 2010; Drinovec et al., 2015). For instance, many studies reporting absorption measurements have been calculating $b_{abs}(\lambda)$ based on Eq. 3 (Weingarter at al., 2003):

$$b_{abs}(\lambda) = \frac{b_{atn}(\lambda)}{C \cdot R(ATN(\lambda))} \qquad \textbf{Eq. 3}$$

where $C$ is the optical enhancement factor due to multiple scattering within the filter medium and $R(ATN(\lambda))$ describes nonlinearities caused by the particles loaded on the filter. Other absorption monitor manufacturers are using different approaches for deriving $b_{abs}(\lambda)$, which can be found in sections 2.2.2 and 2.2.3 for the instruments used in this study.

The equivalent black carbon ($eBC$) mass concentration (expressed in $\mu g\ m^{-3}$) can be calculated based on 880 nm wavelength $b_{atn}(\lambda)$ (Ramachandran and Rajesh, 2007), using either Eq. 4 or 5,

$$eBC = \frac{b_{atn}(880nm)}{\sigma_{atn}(880nm)} \qquad \textbf{Eq. 4}$$

$$eBC = \frac{b_{abs}(880nm)}{MAC(880nm)} \qquad \textbf{Eq. 5}$$

where $\sigma_{atn}(\lambda)$ is the mass attenuation cross-section and $MAC$, the mass absorption cross-section. Table 2 summarizes $C$ and $\sigma_{atn}(\lambda)$ factors used for each instrument in this study. Based on these two parameters MAC can also be calculated by combining Eq. 3, 4 and 5. In this work, the term $eBC$ was chosen instead of BC (Petzold et al., 2013) to stress that BC mass concentration is calculated from optical measurements.

Factor $C$ is considered to be constant during each campaign as it is, relevant to the filter tape only, while $R$ is unity for an unloaded filter and reduces when particles are deposited onto the filter (Weingarter at al., 2003). The filter strip of the miniaturized instruments evaluated in this study is changed manually before every flight to keep the attenuation during a single flight below a threshold value of ATN<0.1 to 0.2, above which loading correction is required (Weingartner et al., 2003; Ferrero et al., 2011).

## 2.2.2 Ground-based (reference) instruments (AE33, MAAP)

To overcome the filter loading effect discussed previously, Drinovec et al. (2015) developed the "dual spot" aethalometer (Magee Scientific, model AE33), which uses two sample spots where particles are deposited with different flow rates and one 'blank' spot as reference. The principle idea behind this approach is that any artefact induced by the accumulation of the particles onto the filter will have the same characteristics (i.e., both sample spots are probing the same particles) but the magnitude of saturation on each spot will differ due to the different amount of the sample on each respective spot. By combining the results from both sample spots, the measurements are extrapolated to zero loading and the compensated/corrected $eBC$ mass and light

absorption can be obtained without using any assumptions on the physicochemical properties of the measured particles.

Another approach for reducing the measuring biases in particle absorption coefficient induced by the accumulation of particles collected on the filter sample spot is employed by the Multiangle Absorption Photometer (MAAP) instrument (Thermo

Fisher Scientific), which applies corrections on the measured absorption coefficient based on the sample-laden particles' scattering at different angles (Petzold and Schönlinner, 2004).

In this study, these two commercially available absorption monitors (Magee Scientific - Model AE33; Thermo Scientific Fisher - Multi-Angle Absorption

Photometer Model 5012) were used as a ground-based reference for UAS-based absorption measurements. Nominally MAAP measurements, which have been shown to agree well against other methods (Sheridan et al., 2005), were used after being corrected based on Eq. 6 (Müller et al., 2011).

$$b_{abs}(637) = 1.05 MAC_{BC}^{MAAP} \cdot eBC \qquad \text{Eq. 6}$$

where $b_{abs}(637)$ is the absorption coefficient at 637 nm (expressed in Mm$^{-1}$), $MAC_{BC}^{MAAP}$ the specific mass absorption coefficient of black carbon proposed by the MAAP manufacturer equal to 6.6 m$^2$ g$^{-1}$ (Petzold and Schönlinner, 2004) and $eBC$ the equivalent mass concentration of black carbon reported by the instrument (in μg m$^{-3}$). Equation 6 assumes that the MAAP operates at a nominal wavelength of 637 nm as

measured by Muller et al. (2011), and not at 670 nm, as proposed by the manufacturer. The absorption coefficient at wavelengths different than 637 nm was calculated based on the Ångström law (Eq. 7).

$$\tau(\lambda) = \tau(\lambda_0) \left( \frac{\lambda}{\lambda_0} \right)^{-\alpha} \qquad \text{Eq. 7}$$

where $\tau(\lambda)$ and $\tau(\lambda_0)$ are the calculated and reference absorption parameters, respectively

and $\alpha$ is the absorption Ångström exponent (AAE). The reported $eBC$ measurements of AE33 were used to calculate $b_{atn}(\lambda)$ and $b_{abs}(\lambda)$ based on Eq. 3 and 4 and using values of mass attenuation cross-section and optical enhancement factor reported in the literature (Table 2). In this work, the absorption coefficient calculated by the AE33 will be scaled to match measurements from MAAP. For the MAAP instrument, the

reference absorption ($\lambda_0$) is at 637 nm, as suggested by Eq. 6. The Ångström exponent was calculated by linear regression of the natural logarithm of the seven wavelength absorption coefficients measured by AE33 (370, 470, 520, 590, 660, 880 and 950 nm) and used for extrapolating into shorter and longer wavelengths of the absorption coefficients measured by the MAAP. Loading correction was not applied to the AE33

measurements as it incorporates a loading compensation measurement scheme (Drinovec et al., 2015).

The AE33 was always operated at a 1 min time resolution; the MAAP operated at a 30 min time resolution during the Athens campaign and at a higher (2 min) time resolution during the Cyprus campaign.


### 2.2.3 Miniature Absorption Monitors (AE51, DWP, STAP)

Three miniaturized instruments having optimal specifications to fly onboard UAS were evaluated. They consist of 1) a single wavelength commercially available

absorption monitor (Aethlabs, Model AE51), 2) a Dual Wavelength Prototype (DWP) monitor based on the AE51 concept, and 3) a Single channel Tricolor Absorption Photometer (STAP; Brechtel Inc - Model 9406). These three instruments will be referred to as AE51, DWP and STAP, respectively, in the following sections. Table 3 summarizes the characteristics of each monitor.

The AE51 is the lightest instrument (280 g) which is a major asset for small UAS observations. On the other hand, due to a relatively low air sampling flow rate (0.1-0.2 L min$^{-1}$ set by the user), it may lack sensitivity for low concentrations of absorbing aerosols which can be an issue when investigating the low amounts of aerosols usually met aloft. The two other instruments (DWP and STAP) have higher flow rates (2 and 1.3 L min$^{-1}$, respectively) which may improve sensitivity for low concentrations. These two instruments also have the potential to derive additional information regarding absorbing material (other than black carbon) using the Aethalometer model reported by Sandradewi et al. (2008). On the other hand, they are significantly heavier (660 g and 1.1 kg for STAP and DWP, respectively) which may represent a major constrain for small UAS operations. The DWP has been constructed as a modification of the AE51, by placing an additional light source, emitting at 370 nm. Additionally, the sampling flow rate has been increased to 2 L min$^{-1}$, by replacing the original AE51 pump, with an external whose flow rate is controlled by a critical orifice. The external pump resulted in additional weight to DWP. In order to assess the possible impact of changes in relative humidity on the attenuation measurements, a second DWP monitor was installed in series behind the one which is been evaluated here. The hypothesis here is that both DWP should be similarly affected by artifacts induced by water absorption/desorption onto the filter strips. An underlying assumption is that both monitors were operating under the same temperature. Under normal (dry) conditions, the second DWP should always report zero concentrations.

The STAP, formerly named ABS (see Bates et al., 2013) has been manufactured following the design of the Particle Soot Absorption Photometer (PSAP; Bond et al., 1999), except that the detection electronics have been completely redesigned to significantly improve signal-to-noise and provide a detection limit of ~0.2 Mm$^{-1}$. Light from three LED sources with wavelengths centered at 445, 515 and 633 nm (Table 3) is alternatively transmitted through glass windows with 50 Hz frequency. The diffused light, which is transmitted through two filter-holding spots that typically carry glass fiber filters, is continuously monitored by two photodetectors. One filter spot is only loaded with the sample aerosol while the other remains sample-free, acting as a reference. The highest measurement rate achieved is 1 Hz. The glass fiber filters minimize light from being transmitted in the forward direction (forward scattering) thus reducing the bias due to scattering by the collected aerosol, while they allow the sampled particles to be embedded within the filter, integrating them in the optically diffusive environment. A laminar flow element is used to measure the sample volumetric flow rate in real-time and an on-board software automatically controls the small integrated vacuum pump to maintain a constant sample volume flow independent of the UAV altitude. The sample flow is dried to eliminate artifacts due to water uptake by the filters.

Calculated absorption from the 3 miniature instruments was derived directly from the sample and reference signals, using Eq 1, 2 and 3 without taking into account the

computed *eBC* or $b_{atn}(\lambda)$ reported by the instruments. For AE51 and DWP, the difference between the calculated and reported absorption values was 0.01% or less. The $b_{atn}(\lambda)$ reported by STAP was initially processed with a 60 s moving average which was deemed too long. To address that issue, a custom-made moving average was applied to the raw (1 Hz time resolution) $b_{abs}(\lambda)$ signal in order to reduce the signal-to-noise ratio (more details in Section 4). Furthermore, this custom moving average allowed a more accurate determination of $b_{abs}(370)$ and $b_{abs}(880)$ based on Eq. 7 for STAP. The STAP manufacturer suggests conversion from $b_{atn}(\lambda)$ to $b_{abs}(\lambda)$ based on Eq. 8 (Ogren et al., 2010), which also accounts for loading artifacts. This conversion has been applied explicitly on STAP measurements instead of Eq. 3 (which has been applied to other miniature absorption monitors).

$$b_{abs}(\lambda) = \frac{0.85 b_{atn}(\lambda)}{1.22(1.0796\dfrac{I(t)}{I_{wf}} + 0.71)} \qquad \text{Eq. 8}$$

Here, *I(t)* is the attenuation at a given time (t) and $I_{wf}$ the measured attenuation of a clean and new filter under particle-free air.

## 3. Sampling Sites

Sampling was conducted at two contrasting locations in the Eastern Mediterranean basin; an urban site (Athens, Greece) for a 1-week intensive period starting from 14 January 2016 and a background location in Cyprus for a 1-month intensive campaign in April 2016.

### 3.1 The Athens campaign

In the framework of the European project ACTRIS 2 (Aerosols, Clouds, and Trace Gases Research Infrastructure), three miniaturized absorption instruments were tested and intercompared for a period of one week (14-21 January 2016) onboard a multicopter over Athens, a city highly impacted by strong UV absorbing domestic heating biomass burning aerosols during winter (Florou et al., 2017; Fourtziou et al., 2017). Flights were conducted at Lofos Nymphon (37°58'19.68"N - 23°43'5.32"E) situated at the historical center of Athens, a metropolitan area of more than 4,000,000 inhabitants. Lofos Nymphon is a rock plateau inside a small forested area (Fig. 2), at a 50 m elevation from its surroundings. Traffic roads, marked with red lines in Fig. 2, are located westerly of the site, the closest of which is 150 m away from the measurement site. In order to comply with air space restrictions made by the Hellenic civil aviation authorities at Lofos Nymphon, the multicopter, described in detail in Section 2.1.3, was selected for its capacity to take-off and land vertically.

A total of 26 flights were performed during periods without precipitation or strong winds. Each flight lasted for 15min and reached as high as 1 km a.s.l. in altitude, a limit set by the Hellenic civil aviation authorities.

During this campaign, the flight plan has been elaborated as the following: two early morning flights were performed at an interval of c.a. one hour starting at sunrise (05:00 UTC) to investigate the stratification of the atmosphere (boundary layer, low free troposphere). Two late afternoon flights ending approximately at sunset (16:00 UTC) were performed to investigate the vertical mixing of urban emissions in the atmospheric

column. On 19 January 2016, intensive (hourly) flights were performed to investigate the impact of the diurnal development of the boundary layer on the vertical distribution of absorbing aerosols. These flights are further discussed in Section 7.

Due to payload restrictions (2 kg maximum for scientific instrumentation and another 2 kg payload for the batteries, dryer, and inlet), not all the miniature monitors could fly simultaneously on board the multicopter. The monitors that could not fly, were operated at the co-located National Observatory monitoring station at Lofos Nymphon, together with two commercially available instruments (AE33; MAAP). In addition, the absorption monitor on board the multi-copter was set to measure at ground level for 2-3 min before and after each flight for a direct comparison against ground-based instruments.

### 3.2 The Cyprus campaign

In the framework of the European project BACCHUS (Impact of Biogenic versus Anthropogenic emissions on Clouds and Climate; towards a Holistic UnderStanding) a 1-month campaign (30 March - 28 April 2016) was performed at the Cyprus Atmospheric Observatory (CAO, 35° 2'17.97"N - 33° 3'28.50"E), a remote regional background site at the Agia Marina Xyliatou in Cyprus.

Vertical profiles of aerosol absorption were performed above a dedicated UAS airfield (35° 5'41.93"N - 33° 4'54.26"E) located at approximately 7 km north of the CAO (Fig. 2). The airfield, shown in Fig. 1, is associated with a 500 m radius (in the x-y plane) UAS airspace and an additional 500 m radius buffer zone, yielding a total of 1 km radius flight zone granted by the Cypriot civil aviation authorities and extending up to a height of approximately 2.4 km a.g.l. (2.7 km a.s.l.).

In this work, only the absorption measurements will be examined corresponding to a total of 17 flights performed with the Skywalker X8 and 6 flights with the Cruiser. The UAS flight strategy was designed to characterize the boundary layer and free troposphere with respect to aerosol absorption, number size distribution, and ice nuclei (IN) concentrations (see Schrod et al., 2017). The typical UAS flight period usually spanned from sunrise (05:00 UTC) to 09:00 UTC. Two types of fixed-wing UASs were used during this campaign; two Skywalker UAS (Model X8) and one Cruiser UAS (see section 2.1). Skywalker X8 flights typically lasted 30 min, while each Cruiser flight lasted between 1-1.5 h. Vertical profiles were performed almost on a daily basis provided meteorological conditions were favorable and engaged a team of eight persons (two pilots, two ground control station operators, two electronic/mechanical engineers and two scientific staff for the operation of the miniaturized instruments).

Ground-based absorption measurements were conducted in parallel at CAO using two commercially available instruments (AE33 and MAAP, see section 2.2.2). CAO is located 6.74 km southerly and at a 200 m elevation from the airfield (Fig. 2). Because of no significant local contamination sources in the surrounding area (Kleanthous et al., 2014; Pikridas et al., 2018), it has been assumed that the atmospheric composition at CAO and the UAS airfield were similar, allowing a direct comparison between the ground and airborne measurements. During this campaign, regional dust transport originating from Africa was identified on two occasions: 9 and 20 April 2016 (Schrod et al., 2017).

During both campaigns lidar measurements at 532 nm from the EARLINET PollyXT-NOA system, described by Engelmann et al., (2016), were used to detect the PBL depth. During the Athens campaign, measurements were collocated with the in-situ measurements described in Section 2.2.2. During the Cyprus campaign, the PollyXT measurements were located 21 km east of the ground-based measurements. Nevertheless, spatiotemporal homogeneity has been observed between the two sites for that specific period (Mamali et al., 2018; Marinou et al., 2018). The PollyXT lidar quick looks from both campaigns can be found online (http://polly.tropos.de).

## 4. Data exploitation: Improvement of the Optimized Noise-reduction Averaging (ONA) Smoothing Algorithm

The three miniature absorption monitors were set to sample at a rate of 1 Hz. However, all measurements were subjected to non-negligible instrumental noise (defined as one single standard deviation of the absorption coefficient) making the data exploitation for short time intervals challenging. The use of a standard averaging method (average, rolling average, least-squares fit) would require setting a fixed time step during which all measurements will be averaged regardless of the signal-to-noise ratio. This will reduce noise but may compromise the need for high time (spatial) resolution required for UAS-based vertical profile measurements. Instead, Hagler et al. (2011) proposed a method where the averaging step is not defined by the time but is based on the measured attenuation. In that method, named Optimized Noise-reduction Averaging (ONA), $dATN(\lambda)/dt$ should only be positive or zero (but not negative, an assumption which is valid in our case without any fresh volatile sample fraction). As a result, for a predefined configuration (sample volume, sample spot area), the same averaging attenuation step ($\Delta$ATN) will require more data points to be averaged during periods with low atmospheric concentrations (i.e lower time resolution) compared to periods with high atmospheric concentrations. Therefore, using ONA, the averaging time step is dynamically set to be inversely proportional to the sampled concentration (see also Eq. 2), resulting in a fixed signal-to-noise ratio. Since the method is based on attenuation changes, it can only be applied to individual spots, where the sample is accumulated, in a continuous monitor or an individual filter in semi-continuous monitors such as the miniature absorption monitors investigated in this work.

The algorithm proposed by Hagler et al. (2011) results in an integrated-like (fragmented) data structure which lowers the vertical resolution of our UAS-based absorption measurements significantly (blue dots in Fig. 3). To cope with this issue, an improvement of the ONA algorithm is proposed here. A moving average is implemented instead of the one applied in the ONA algorithm, resulting in a more continuous-like data structure and improved vertical resolution (red dots in Fig. 3). If more than one wavelength is monitored, then the improved ONA algorithm can be applied to each of the wavelengths but based on the same attenuation, in order to produce comparable averaging results. The same strategy can be applied to external datasets for comparison purposes, provided they are produced or conditioned to have the exact same time resolution.

The flow diagram of the proposed improved ONA algorithm is presented in Suppl. Fig. 1. A link to the actual code is also provided, via a file-sharing portal, in the

supplement. The user supplies attenuation, and instrument response (e.g. *eBC* mass, $b_{abs}$, or an external measurement) as time series along with the desired attenuation step (ΔATN). The calculated time interval includes attenuation values in the range [-0.5×ΔATN, + 0.5×ΔATN] centered at the selected data point. If the attenuation change of a data point is greater by 0.5×ΔATN with respect to its neighbors, then this data point will not be smoothed. The time interval is limited to correspond to only one sample spot. The same averaging times can be then applied to the remaining monitored wavelengths if any. Discrepancies could arise when abrupt concentration gradients are sampled e.g monitoring the vertical profile of a polluted boundary layer followed by clean air masses. In this case, the rate of attenuation change will decrease, since the air mass contains less absorbing aerosol. If the concentration gradient between the two layers is large enough the algorithm may lead to a fictitious shift of the boundary layer height because more data points from the clean air mass than the polluted boundary layer will be accounted for in the average. The discrepancy is solved if weights inversely proportional to the number of data points are used for the average before (-0.5×ΔATN) and after (+ 0.5×ΔATN) the sample point to be examined. The improved ONA algorithm incorporates filters that cope with this problem. Erroneous results may also arise from outliers in the time series, especially if small ΔATN is applied or if the time series is over smoothed. An example of over smoothing is shown in Fig.3 (green line). For all the reasons discussed above, it is advised to examine the result using different ΔATN and against the raw input.

High ΔATN values will reduce noise but reduce the time (vertical) resolution. A ΔATN equal to 0.01, 0.03, 0.03 is suggested for AE51, DWP and STAP, respectively, and these values take into account the air face velocity set for each instrument. Vertical profile case studies are therefore discussed later in Section 7 with the above-proposed attenuation steps. Note that Hagler et al. (2011) suggests a higher ΔATN, equal to 0.05, for all monitors regardless of individual face velocity.

## 5. Quality Assurance

Despite that all the available methods have the scope of reporting the mass concentration of BC, discrepancies between the different techniques or even instruments that are based on the same operating principles have been reported (eg., Watson et al., 2005; Slowik et al., 2007; Müller et al., 2011). These discrepancies are not only attributed to the different measurement techniques/instruments used but also to the large variability of the physicochemical properties of atmospheric or laboratory-generated carbonaceous particles. For instance, the optical properties of carbonaceous particles depend on their size and morphology (Bond and Bergstrom, 2006; García Fernández et al., 2015), on their mixing state and/or coating thickness with other atmospheric relevant species, including sulfate, water, organic or dust (Lack and Cappa, 2010; Shiraiwa et al., 2010; Lack et al., 2014; Liu et al., 2015; Zhang et al., 2015, 2018) As a result, aerosol absorption measurements need to be associated with a comprehensive understanding of the methods and uncertainties associated with each instrument and how they have been operating and operated in the field. Condensation or volatilization of water on the filter spot of the miniature sensors can greatly affect absorption measurements (Hale and Querry, 1972; Düsing et al., 2019). In order to minimize this artifact, a custom-built (lightweight) silica-gel dryer was installed at the

inlet of each miniature sensor and regenerated before each flight. Each sensor operated with its own respective inlet and dryer during both campaigns and even when two sensors were airborne simultaneously in one UAS. However, to reduce weight, no size-selective inlet was employed. Ground-based sensors were similarly configured, at least when UAS flights were ongoing.

In the following sections, the level of agreement, at the 95% confidence interval (CI) between standard (rack) size absorption monitors and miniature absorption sensors will be evaluated using an adaptation of the standard Student's t-test (Welch, 1947) that accounts for samples with unequal variances and unequal sample sizes. Because the test is valid only for normal distributions the datasets to be compared are transformed (e.g.

Box Cox transformation) and tested by an F-test (Box, 1953) to satisfy this assumption.

### 5.1 Aerosol Absorption derived by AE33 and MAAP

During the Athens campaign AE33 and MAAP showed excellent correlation ($R^2$=0.98, N=381) with respect to the *eBC* mass concentration trend at a 30 min time

resolution (Fig. 4). However, AE33 reported higher eBC by 20±11% compared to MAAP, and higher absorption coefficient at 370, 637 and 880 nm of more than a factor 2. Each of these differences is statistically significant at the 95% CI. During the Cyprus campaign, both monitors also showed a very good correlation ($R^2$=0.89, N=1434) at a 30 min time resolution. However, similar to the Athens campaign, AE33 showed *eBC*

mass concentration higher by 13±5% compared to MAAP, and higher absorption coefficient at 370, 637 and 880 nm by almost a factor 2, which was also significant at 95% CI. It is noted that for both campaigns the absorption coefficient has been derived from *eBC* for both instruments. The large difference observed concerning the absorption coefficient is due to the different generic *MAC* values applied to each

instrument. As an example, the *MAC* value employed by MAAP is equal to 6.6 $m^2$ $g^{-1}$ at 637 nm (Table 4), while the *MAC(637)* calculated for AE33 is equal to 10.7 $m^2$ $g^{-1}$. For both campaigns, the comparison of *eBC* and the absorption coefficient at 370 and 880 nm is shown in Fig. 4, and for the absorption coefficient at 637 nm at Suppl. Fig. 2.

Drinovec et al. (2015) suggested that AE33 could overestimate *eBC* up to approximately 7% when compared to MAAP. Müller et al. (2011) calculated the absorption coefficient at 637 nm of single spot aethalometers measuring ambient air and showed that it can be up to 60±20% overestimated when compared to MAAP. Finally, MAAP has been reported to underestimate *eBC* in polluted environments

(Hyvärinen et al., 2013) when the measured *eBC* concentration exceeds 3 $\mu g$ $m^{-3}$. Table 4 summarizes the results from both campaigns (illustrated in Fig 4). This comparison suggests that AE33 and MAAP exhibit a better match with respect to *eBC* mass rather than with the absorption coefficient.

In the comparison presented above, MAAP was chosen as the reference instrument

because it has been shown to exhibit good agreement against ambient absorption methods (Sheridan et al., 2005) that do not require correction schemes (e.g. photoacoustic spectrometers) and because its unit-to-unit variability reported to be small (approximately 5%; Müller et al., 2011). However, MAAP monitors absorption at a single wavelength and samples at lower temporal resolution than the one desired

for this study (30 min in the Athens campaign and 2 min in the Cyprus campaign).

In the following sections, we investigate how measurements from miniature attenuation monitors relate to the commercial ones discussed in this section. AE33 is always utilized as a reference because of its high temporal resolution (1 min). For this purpose, AE33 results are first scaled to match those of MAAP, to approximate, at least on average, the suggested "reference" values taking advantage of the excellent trend agreement between these two instruments. The $eBC$ by the AE33 was consequently decreased by 20% and 13%. The difference in the scaling factor between the two campaigns is attributed to instrument variability since two different pairs (of AE33 and MAAP) were employed in each campaign and to the different aerosol sampled, fresh vs aged during the Athens and Cyprus campaigns, respectively. Consequently, $b_{abs}(370)$ was decreased by a factor of 2.4 and 1.93, and $b_{abs}(880)$ was decreased by a factor of 2.2 and 1.83 during the Athens and Cyprus campaigns, respectively.

**5.2 UAS-based absorption measurements**

The loading correction term in Eq. 3 was neglected in our study, assuming a value equal to unity when attenuation was low. It is noted that currently, most loading correction schemes are applied to continuous monitors that change sample spots automatically. Attenuation of AE51, provided by the instrument never exceeded 0.01 during the Athens campaign due to the combination of low sampling flow rate and limited sampling times (approximately 15 min) of each flight. During the Cyprus campaign, it reached up to 0.02 because sampling time was higher (1-1.5 h) despite the lower measured $eBC$ concentrations. Because of its higher sampling flow rate, the attenuation of DWP at 880 nm exceeded 0.15, five times in each of the two campaigns. In order to examine whether measurements by DWP exceeding attenuation of 0.1 were significantly affected by the filter loading effect, a comparison with respect to $b_{abs}(880)$ was conducted against both AE51 and AE33. The comparison results, shown in the supplementary material (Suppl. Fig. 3), support the assumption of a loading correction ($R$) equal to unity was valid during both campaigns (as already discussed by Weingartner et al., 2003).

As discussed in Section 2.2.3, the DWP configuration consisted of two monitors sampling in series, in order to assess the possible impact of changes in relative humidity on the attenuation measurements. Under dry conditions, the second DWP should always report zero concentrations; this was the case during the Athens campaign with the exception of one flight performed on the 15$^{th}$ January 2016 when the silica gel dryer was removed. During this flight, the second DWP provided attenuation measurements deviating from zero, as high as 30 Mm$^{-1}$ at 880 nm, suggesting that the first DWP measurements may also have been affected by sampling bias during this particular flight (Suppl. Fig. 4). Recently, Düsing et al. (2019) evaluated the discrepancy due to RH gradients of STAP to be 10.08 Mm$^{-1}$ s$^{-1}$ for every 1% change in RH.

**6. Comparison of miniature attenuation monitors against reference instruments**

Since most of the commercially available sensors provide BC readings (instead of absorption like STAP), we have decided to extend our absorption intercomparison to $eBC$. Despite BC being the most absorbing material in ambient air, other components, such as brown carbon and dust could also contribute to absorption especially at shorter wavelengths (Andreae and Gelencsér, 2006). In addition to $eBC$, aerosol absorption

coefficients at 370 and 880 nm were also selected because two of the three miniaturized sensors measured at least at one of those wavelengths (see Table 3). Extrapolation based on Ångström law (Eq. 7) was applied for STAP that did not measure at these two specific wavelengths using as a base the 445 nm and the 633 nm channels to convert to 370 nm and 880 nm, respectively.

**6.1 Overview of the temporal and diurnal variability of ground-based *eBC* during the Athens and Cyprus campaigns**

During the Athens campaign, the average *eBC* concentration determined by AE33 was $1.5\pm2.1$ μg m$^{-3}$, ranging from 0.3 to 15 μg m$^{-3}$. The presence of BC from biomass burning (BC$_{bb}$), was identified and quantified throughout the campaign (Suppl. Fig. 5), using the Sandradewi et al. (2008) model, but never exceeded 20% of the total *eBC* during daytime (05:00-15:00). During the nighttime, BC$_{bb}$ concentration was always elevated, reaching 40-60% of the total *eBC* that typically remained below 2 μg m$^{-3}$. On two occasions (14 Jan 16:00 – 15 Jan 05:00 and 21 Jan 15:00 – 22 Jan 00:00) *eBC* exceeded 5 μg m$^{-3}$ for several hours dominated by BC$_{bb}$. On average, BC$_{bb}$ was identified from 16:00 UTC till 04:00 UTC of the following day and was more prominent during the periods featuring a low boundary layer and the need for heating due to low temperatures. Similar behavior attributed to biomass burning aerosol has been reported previously in Athens (Florou et al., 2017; Fourtziou et al., 2017) and other major Greek cities (Petrakakis et al., 2013; Pikridas et al., 2013). BC related to fossil fuel also exhibited a distinct diurnal pattern that included two maxima (Suppl. Fig 5). The first was observed approximately 06:00 UTC that was attributed to the rush hour traffic period and the second in late afternoon/evening (after 16:00 UTC) simultaneously with the period when biomass burning related BC was observed. Increased biomass burning, especially during nighttime for domestic heating purposes, due to the economic crisis in Greece, has been reported for another major Greek city (Saffari et al., 2013).

During the Cyprus campaign, *eBC* measured by AE33 did not exceed 2 μg m$^{-3}$ and most of the time it was found below 0.8 μg m$^{-3}$. The highest hourly concentration (1.9 μg m$^{-3}$) was observed on the 10 April 2016 (Suppl. Fig. 6) when the site was influenced by air masses from N. Africa, and the lowest (<0.1 μg m$^{-3}$) on the 12 and 14 of April 2016. During the Cyprus campaign, dust transport from the Saharan desert was identified on 3 occasions (7-10, 15-17 and 21-27 April 2016) based on combined information from i) elevated coarse-mode particulate matter concentrations measured by a tapered element oscillating microbalance (Thermo model 1400a), ii) aerosol spectral properties of the entire atmospheric column measured by sun photometry iii) back-trajectory analysis and iv) satellite pictures (MODIS AOD product). The diurnal pattern of *eBC* during the Cyprus campaign was relatively flat as expected in a remote background site, characterized by an almost invariable concentration approximately at 0.4 μg m$^{-3}$ (campaign average equal to $0.39\pm0.24$ μg m$^{-3}$).

**6.2 Ground-based intercomparison of aerosol absorption**

During the Athens campaign, each miniature sensor not performing vertical profiling was operating at ground level in parallel with AE33 and MAAP, allowing a direct comparison. Additionally, the miniature sensors on-board the multi-copter were

measuring at ground level (2-3 min) before take-off and after landing. It is noted that the same setup (sampling lines, diffusion dryer) was utilized whether the miniature samples were mounted in the UAS platform or not. Based on the combination of these datasets resampled to 1 min (the time resolution of AE33), DWP exhibited good correlation, with respect to *eBC* against AE33 ($R^2$=0.90, slope=0.93, N=417) shown in Fig. 5a, while the AE51 produced slightly poorer correlation ($R^2$=0.76, slope=0.94, N=125) (see Table 4). One possible explanation is the lower signal-to-noise ratio of AE51. Both monitors measured *eBC* concentrations lower by 6-7% compared to the reference measurements. This difference is not statistically significant, at the 95% CI, for both DWP and AE51. STAP does not report *eBC* mass concentration and was excluded from this comparison for that purpose.

With respect to $b_{abs}(\lambda)$ at 370 and 880 nm, both STAP and DWP showed good correlation (At 370 nm : $R^2$=0.89 and 0.87, N=519 and 417 for STAP and DWP, respectively; At 880 nm : $R^2$=0.88 and 0.9, N=519 and 417 for STAP and DWP, respectively) against AE33, while the correlation with AE51 was slightly poorer ($R^2$=0.76, N=125) at 880 nm (Fig. 5c).

However, DWP overestimated $b_{abs}(880)$ by 29±20% (significant at 95% CI) compared to the corresponding reference measurements, even though the *eBC* mass, calculated from the same wavelength, was underestimated by 7%. Similar to DWP, AE51 overestimated $b_{abs}(880)$ by 30±12% even though *eBC* mass was underestimated by 6%. This difference was statistically significant at 95% CI but only marginally (p-value equal to 0.049). Both DWP and AE51 share the same $\sigma_{atn}$ and *C* values (Table 4). For both instruments, a generic *MAC(880)* value equal to 6.1 $m^2$ $g^{-1}$ is applied to convert *eBC* to $b_{abs}$, instead of 7.8 $m^2$ $g^{-1}$ used by AE33 at the same wavelength. However, both miniature sensors underestimate with respect to *eBC* but at the same time overestimate with respect to the absorption coefficient mainly due to the higher correction factor applied to the AE33 measurements concerning the latter (approximately a factor of 2) compared to the former (≈20%) to match those of MAAP as discussed in Section 5.1.

STAP was found to overestimate $b_{abs}(880)$ by 6±8.5% and underestimate $b_{abs}(370)$ by 7±7%. Both differences were not significant at 95% CI. During a laboratory comparison (Müller et al., 2011) reported that a continuous single spot aethalometer (Magee Model AE31) overestimated $b_{abs}$ compared to MAAP by 37-60% at 660 nm. The same study also reported underestimation of the absorption coefficient at 650 and 585 nm against MAAP compared to the PSAP (the rack-mounted equivalent of STAP) by 1-14%. These laboratory comparison results are similar to those reported in this study (AE51 overestimates and STAP underestimates the absorption coefficient by a similar extent against the reference).

The miniature sensors intercompared during the Athens campaign exhibit better agreement with respect to the parameter they report. Concerning AE51 and DWP, this parameter was *eBC* concentration, which was within 10%, rather than the absorption coefficient, suggesting that the absorption coefficient should be preferentially calculated based on a single set of wavelength-dependent MAC values (Eq. 5) instead if these are known or can be calculated. On the other hand, STAP that does not report *eBC* but $b_{abs}$ exhibited good agreement, within 10%, against the reference on that property. On average the calculated AAE of DWP and STAP is underestimated by 13% and 12% respectively against that of AE33.

During the Cyprus campaign, aerosol absorption was also monitored at the ground by an AE33 and a MAAP located at CAO, approximately 7 km away and at 200 m higher elevation from the UAS airfield. Only DWP and AE51 were used on UAS during this campaign. Assuming homogeneity between the two sites, a direct comparison was conducted between ground and UAS measurements.

The comparison results, shown in Fig. 6, indicate that the correlation between the ground measurements and UAS (AE51 and DWP) measurements led to less satisfactory results compared to the Athens campaign (see also Table 4). The correlation between AE33 and DWP was still acceptable ($R^2$=0.71; N=91) with respect to *eBC* and the absorption coefficient at 370 and 880 nm at 1 min time resolution. But the correlation between AE33 and AE51 was found poor ($R^2$=0.32, N=48) with respect to both *eBC* and $b_{abs}(880)$.

The atmospheric concentration of absorbing material (*eBC* measurements) was found on average 4 times lower in Cyprus (mean of 0.39±0.24 µg m$^{-3}$) compared to Athens (mean of 1.5±2.1 µg m$^{-3}$). Additionally, the range of atmospheric concentrations was also reduced by a factor of 6 in Cyprus (maximum hourly averaged *eBC* was 1.9 µg m$^{-3}$) compared to Athens (maximum hourly averaged *eBC* was 12.2 µg m$^{-3}$), leading to less favorable conditions for direct instrument-by-instrument comparisons due to the smaller range of comparison data. These conditions had a direct impact on the uncertainty related to the measurement agreement between the AE33 and the miniature monitors. During the Cyprus campaign, the uncertainty was always greater than the respective of the Athens campaign. As an example during the Cyprus campaign, DWP underestimated *eBC* by 6±20% and overestimated $b_{abs}(880)$ by 20±26% (both significant at 95% CI), while during the Athens one the respective numbers were 7±15% and 29±20% (Table 4). The effect was greater concerning AE51, which overestimated *eBC* by 22±52% and $b_{abs}(880)$ by 55±66%, while during the Athens campaign the respective numbers were 6±9% and 30±12% (Table 4). Due to the large uncertainty exhibited by AE51, the null hypothesis that the population mean of the reference instrument (AE33) and of AE51 are different was not met. Hence, the reported differences are not significant at the 95% CI. It is unclear whether the absorbing properties of the sampled aerosol (fresh at Athens and aged in Cyprus) had any effect on this comparison.

**6.3 On flight intercomparison of aerosol absorption**

During flights, vibrations, as well as strong gradients of pressure, temperature, and RH may affect the performance of the miniature sensors. In order not to surpass the maximum take-off weight of the multicopter used during the Athens campaign, STAP and DWP did not fly simultaneously. However, the lower weight of AE51 enabled on-flight cross-comparison with DWP and STAP, respectively during 8 flights of the Athens campaign. The correlation of AE51 airborne with both DWP and STAP was very good ($R^2$=0.65, N=493 and $R^2$=0.87, N=1875, respectively) provided that the sampled air was dried (Fig. 7) and the dataset post processed with a noise-reducing algorithm as suggested in Section 4. Error bars shown in Fig. 7 correspond to one standard error for one-second time resolution. In the case that the algorithm did not average a sampling point with its neighbors, then by default, the standard deviation and standard error were zero, indicated by a lack of an error estimate in Fig. 7. Note that if

no smoothing is applied, the correlation deteriorates sharply ($R^2$=0.01) for either DWP
        or STAP. The ΔATN used for this comparison were 0.01, 0.03, and 0.03 for AE51,
        DWP, and STAP, respectively as suggested in Section 4. STAP is shown to
        underestimate $b_{abs}$ by 12% (significant at 95% CI) compared to AE51 (Fig. 7),
        consistent with the comparison against AE33 discussed in Section 6.2. The very good
correlation (comparison slope = 0.87) between the two when airborne also suggests that
        on average, no significant bias during the flights was present. The difference between
        AE51 and DWP was 8%, which was not significant at 95% CI.

**7. Diurnal Vertical Profiles of Black Carbon above Athens: A case study**

        As part of the Athens campaign, intensive vertical absorption profiles were
performed with the objective to assess the influence of the diurnal development of the
        planetary boundary layer (PBL) on the vertical dispersion of ground-based black carbon
        emissions. UAS-based measurements were conducted for that purpose on the 19
        January at sunrise (05:38 UTC) and were continued on an hourly basis till the PBL
        depth exceeded the maximum height allowed to operate (1 km a.s.l.) approximately at
10:00 UTC. Two additional flights were conducted later on that day; one hour before
        and during sunset (15:38 UTC). The reconstructed vertical distribution of *eBC* based
        on the six ascending vertical profiles from 05:30 till 09:45 (UTC) is shown in Fig. 8,
        complemented by ground measurements during the same day by AE33. The actual
        vertical profiles for the entire day (N=8) are also shown in Fig. 9. We present a very
detailed study of vertical dispersion of ground-based black carbon emissions
        dynamically assessed above a major city. Our results suggest a non-homogeneous
        boundary layer that evolved at a rate of 132 m h$^{-1}$ during 19 January 2016 starting from
        an elevation of 265 m a.s.l. before sunrise. Starting at 05:00 UTC *eBC* increased by a
        factor of 8 at 07:00 UTC. The emission's pattern and the Ångström exponent, calculated
based on AE33 measurements, which was equal to 1.1 when concentrations maximized,
        suggest that this increase in *eBC* was due to local traffic emissions (see also Fig. 8).
        After 10:00 UTC *eBC* remained relatively stable at 1.5 µg m$^{-3}$ (≈5 Mm$^{-1}$ at 880 nm).

        Above the PBL, which was determined by Polly-XT measurements (Baars et al.,
        2008; dashed red lines in Fig. 9), the measured concentration of *eBC* was always lower
than the respective one measured within, by at least 20%. The highest *eBC*
        concentrations above the PBL were observed during sunrise and sunset (first and last
        diurnal profile in Fig. 9) equal to 1.9 and 2.0 µg m$^{-3}$, respectively, which we interpret
        as the residual layer of the previous day in the morning and the newly formed residual
        layer after sunset. The lowest *eBC* concentration in this layer, equal to 0.3 µg m$^{-3}$, was
observed at 06:30 UTC but steadily increased to 0.4, 0.9 and 1.7 µg m$^{-3}$during 07:38,
        08:39 and 09:44 UTC, respectively. Due to flight restrictions, free tropospheric
        measurements could not be monitored after 10:00 UTC. PBL was also identified by
        vertical profiles of potential temperature which are in good agreement with those
        derived by Polly-XT

Before sunrise, our results suggest the presence of a stable boundary layer in contact
        with the ground that has been radiatively cooled; on top of the boundary layer, there is
        a residual layer. As the sun rises, the stable boundary layer's depth increases and
        simultaneously the residual layer is mixed with the free troposphere. On the 19 January
        2016, mixing took place between 05:45-06:30 UTC. The concentration of *eBC* in the

residual layer drops to near zero because the trapped pollutants are now diluted in the free troposphere.

However, the concentration of *eBC* above the boundary layer exhibited an increasing trend suggesting either convection of pollutants from the PBL or advection of regionally transported PM involving absorbing material that did not intrude the PBL. During the period when absorbing material was directly emitted from the ground and the boundary layer height increased (from 05:30-08:30 UTC), *eBC* dispersion inside the PBL was not homogeneous but was gradually decreasing with increasing altitude. The effect is more evident when emissions from the ground exhibited an increasing trend (approximately from 06:30-07:40 UTC). Once ground emissions reached their minimum and the PBL stabilized, the concentration inside the PBL became homogeneous (from 10:00 UTC till sunset). During sunset, stratification of a new stable boundary layer was observed and on top of it a new residual layer forming.

The vertical absorption distribution was reconstructed based on the absorption profiles shown in Fig.8, on 19 January 2016 between 05:34 and 09:36 (UTC) and also shown in Fig. 9 against calculated attenuated backscatter at 1064 nm measured by a PollyXT.

## 8. Conclusions

Two field campaigns were conducted in Athens (Greece) and in CAO (Cyprus) in order to i) study the vertical distribution of aerosol absorption and ii) to evaluate the performance of three miniature absorption sensors in contrasted atmospheric environments against ground-based reference instruments (MAAP and AE33). Measurements were conducted on the ground and air using three different models of UASs. Our results suggest that the absorption monitors used in this work agree better at the parameter they report, which is *eBC* in most cases, rather than the absorption coefficient. This discrepancy is directly related to the generic *MAC* values suggested by the manufacturer of each instrument. In case the absorption coefficient is not directly reported, it should be preferentially calculated based on a single set of wavelength-dependent *MAC* values specific to each site, if these are known or can be calculated, instead of the generic provided by the manufacturer.

The influence of humidity on attenuation measurements was investigated during the Athens campaign, by placing two DWP in series, with the second measuring filtered air from the exhaust of the first. Sample drying minimized the influence of water adsorption/desorption on attenuation measurements.

During January 2016, the miniature sensors sampled urban aerosols at the center of Athens, Greece. On the ground, STAP and DWP followed well the observed variations in the absorption ($R^2 \approx 0.90$) against an AE33, while AE51's performance ($R^2 = 0.76$) was poorer due to low sampling flow rate. STAP was found to overestimate absorption coefficient at 880 nm by 10%, while AE51 and DWP overestimate it by 40% and 30%, respectively. However, with respect to *eBC* mass, the agreement was closer (within 7%). An inflight intercomparison between the lightweight AE51 and either the STAP or DWP was achieved during the Athens campaign. No correlation between the AE51 and STAP or DWP could be achieved for unconditioned high-time resolution (1 Hz) measurements. An improvement of the smoothing algorithm suggested by Hagler et al. (2011) was applied here leading to improved correlations ($R^2 > 0.70$) between miniature

sensors (AE51, DWP and STAP). Based on four UAS flights, DWP and AE51 correlated very well (comparison slope equal to 0.92) with respect to the absorption coefficient at 880 nm ($b_{abs}(880)$), while STAP was found to underestimate $b_{abs}(880)$ by 12% which was consistent with the intercomparison performed at ground level against the AE33.

The Cyprus campaign took place at the Cyprus Atmospheric Observatory, a remote location distant by 7 km from the UAS runway and two of the miniature sensors (DWP and AE51) were evaluated in-flight against ground-based reference instruments, taking advantage of the elevation difference between the two sites. By comparison to the Athens campaign, the correlation of both sensors (against reference instruments) deteriorated because of low atmospheric aerosol concentrations (4 times lower) and reduced atmospheric variability (6 times lower). While DWP showed relatively good correlation (R²=0.71; N=91 data points) and the same level of agreement as during the Athens campaign (6% overestimate), the poor performance of AE51 ($R^2$=0.32, N=91) was attributed to a lack of sensitivity of this sensor operating at a flow rate c.a. 10 times lower compared to DWP.

The overall potential use of miniature aerosol absorption sensor on-board small UAS was illustrated with results of the campaign performed in Athens. During this campaign, the diurnal variability of the vertical distribution (0-1 km a.g.l.) of equivalent Black Carbon was investigated. It was found that *eBC* concentrations are not homogeneous in the boundary layer when it develops (PBL depth increases) and simultaneously absorbing material is emitted at ground level by traffic. Vertical homogeneity of *eBC* is reached in the afternoon when the boundary layer height is stabilized and emissions at the ground are reduced.

Acknowledgments: The two field campaigns (Athens, Cyprus) are a contribution to the ACTRIS2 project that has received funding from the European Union's Horizon 2020 research and innovation program under grant agreement no. 654109. EU FP7 project BACCHUS (project number 603445) is acknowledged for financial support. MV acknowledges support from the DFG-Research Center/Cluster of Excellence "The Ocean in the Earth System-MARUM". MP acknowledges the financial support of the CURE-3AB project (INTERNATIONAL/OTHER/0118/0108) which is co-financed by the European Regional Development Fund and the Republic of Cyprus through the Research and Innovation Foundation. EM acknowledges the financial support of the Deutscher Akademischer Austauschdienst (grant no. 57370121), VA acknowledges the financial support of the European Research Council (grant no. 725698, D-TECT).

## 9. Nomenclature

| Abbreviation | Description |
|---|---|
| AAE | Absorption Ångström exponent |
| ACTRIS | Aerosols, Clouds, and Trace Gases Research Infrastructure |
| a.s.l. | Above sea level |
| ATN | Attenuation |
| $b_{atn}$ | Light attenuation coefficient |
| BACCHUS | Impact of Biogenic versus Anthropogenic emissions on Clouds and Climate; towards a Holistic UnderStanding |
| $b_{abs}$ | Light absorption coefficient |
| BC | Black carbon |
| $BC_{bb}$ | BC related to biomass burning |
| C | Optical enhancement factor |
| CAO | Cyprus atmospheric observatory |
| CI | Confidence interval |
| DWP | Dual-wavelength prototype |
| EARLINET | European Aerosol Research Lidar Network |
| $eBC$ | Equivalent black carbon |
| EC | Elemental carbon |
| GUI | Graphical user interface |
| MAAP | Multiangle Absorption Photometer |
| MAC | Mass absorption cross-section |
| MTOW | Maximum take-off weight |
| ONA | Optimized Noise-reduction Averaging |
| PBL | Planetary boundary layer |
| PSAP | Particle Soot Absorption Photometer |
| R | Filter loading parameter |
| STAP | Single-channel Tri-color Absorption Photometer |
| UAS | Unmanned aerial systems |
| $\alpha$ | Ångström exponent |
| $\lambda$ | Wavelength |
| $\sigma_{atn}$ | Mass attenuation cross-section |

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

**Table 1**. Summary of UAS used during the Athens and Cyprus campaigns. A UAS is considered small if its gross weight is smaller than 25 kg and medium if its gross weight ranges between 25 and 50 kg.

| USRL Fleet of UAS | Type | MTOW[†] | Payload* | Endurance* | Ceiling* | Manufacturer |
|---|---|---|---|---|---|---|
|  Cruiser | Medium Size Fixed Wing | 35 kg | 12 kg | 4 h | 3km a.s.l. | ET-Air |
|  Skywalker X8 | Small Size Fixed Wing | 5 kg | 3 kg | 1 h | 3 km a.s.l. | Skywalker |
|  DJi S1000+ | Small Size Rotary Wing | 11 kg | 4 kg | 30 min | 1 km a.s.l. | DJi |

* UAS characteristics as configured particularly for these studies (BACCHUS and ACTRIS campaigns).
† Maximum take-off weight



**Table 2.** Summary of standardized properties of each attenuation monitor. The term $\lambda$ refers to the wavelength used in nm.

| Instrument name | Manufacturer | Mass attenuation cross section ($m^2\ g^{-1}$) | Optical enhancement factor (C) | Reference |
|---|---|---|---|---|
| AE33 | Magee Scientific | 10730.48/$\lambda$ | 1.57 | Drinovec et al., 2015 |
| AE51 | Magee Scientific | 11000/$\lambda$ | 2.05 | Ferrero et al., 2011 |
| STAP | Brechtel | N/A* | N/A* | Ogren et al., 2010 |
| MAAP | Thermo Scientific | 6.6 at 670 nm | N/A | Petzold and Schönlinner, 2004 |
| Dual Wavelength Prototype (DWP) | Custom made from AE51 | 11000/$\lambda$ | 2.05 | N/A |

\* Equation 7 is used instead


**Table 3**. Characteristics of the miniature absorption instruments.

| Instrument Name | Flowrate (LPM) | Spot Area (m²) | Wavelengths (nm) | Face Velocity (m s⁻¹) | Weight* (g) | Time Response (s) |
|---|---|---|---|---|---|---|
| AE51 | 0.1-0.2 | $7.1 \times 10^{-6}$ | 880 | 0.5 | 280 | 1, 10, 30 |
| DWP | 2 | $7.1 \times 10^{-6}$ | 370, 880 | 4.7 | 1100 | 1 |
| STAP | 1.3 | $17.7 \times 10^{-6}$ | 445, 515,633 | 1.2 | 660 | 1 |

*Refers to the weight of the instrument alone. Dryer and sampling inlet used are not accounted for.








**Table 4**. Results from the comparison of the miniature sensors with ground-based commercial instruments (AE33 and MAAP) shown in Fig. 5 and 6

| | eBC | | $b_{abs}(370)$ $Mm^{-1}$ | | $b_{abs}(880)$ $Mm^{-1}$ | | Ångström Exponent | |
|---|---|---|---|---|---|---|---|---|
| | slope ±95% CI | Quality of fit ($R^2$) | slope ±95% CI | Quality of fit ($R^2$) | slope ±95% CI | Quality of fit ($R^2$) | slope ±95% CI | Quality of fit ($R^2$) |
| | | | | Athens campaign | | | | |
| **AE33** | 1.20±0.11 | 0.98 | 2.45±0.21 | 0.99 | 2.25±0.19 | 0.99 | N/A | N/A |
| **DWP** | 0.93±0.15 | 0.90 | 1.22±0.20 | 0.87 | 1.29±0.20 | 0.90 | 0.87±0.22 | 0.21 |
| **AE51** | 0.94±0.09 | 0.76 | N/A | N/A | 1.30±0.12 | 0.76 | N/A | N/A |
| **STAP** | N/A | N/A | 0.93±0.07 | 0.89 | 1.06±0.08 | 0.88 | 0.88±0.17 | 0.27 |
| | | | | Cyprus campaign | | | | |
| **AE33** | 1.13±0.05 | 0.89 | 1.93±0.09 | 0.88 | 1.83±0.08 | 0.89 | N/A | N/A |
| **DWP** | 0.94±0.20 | 0.71 | 0.83±0.18 | 0.68 | 1.20±0.26 | 0.71 | 0.44±0.28 | 0.1 |
| **AE51** | 1.22±0.52 | 0.32 | N/A | N/A | 1.55±0.66 | 0.32 | N/A | N/A |






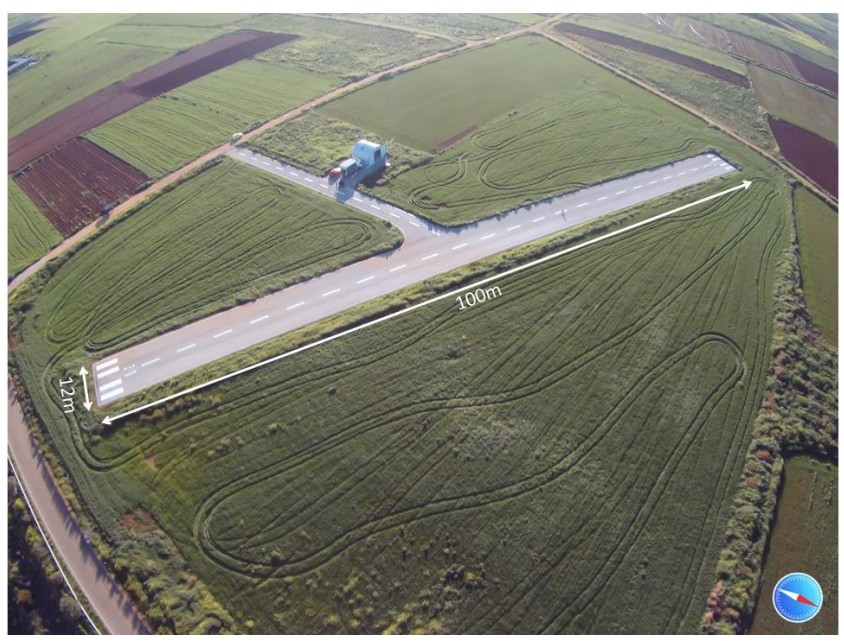

**Figure 1**: Aerial view of the Orounda runway in Cyprus.



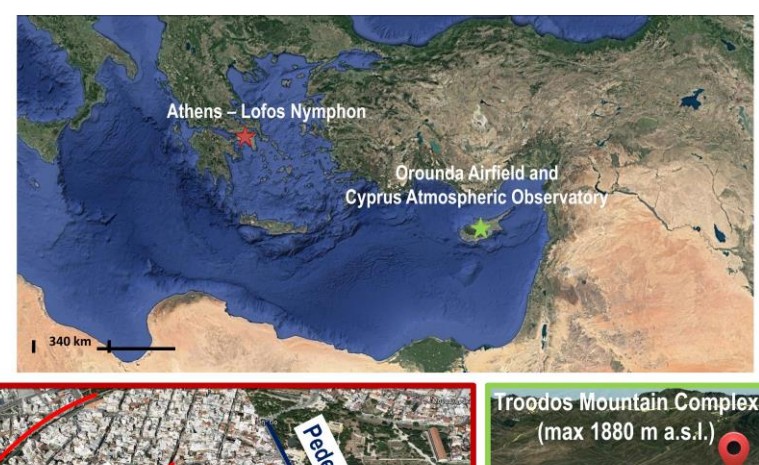

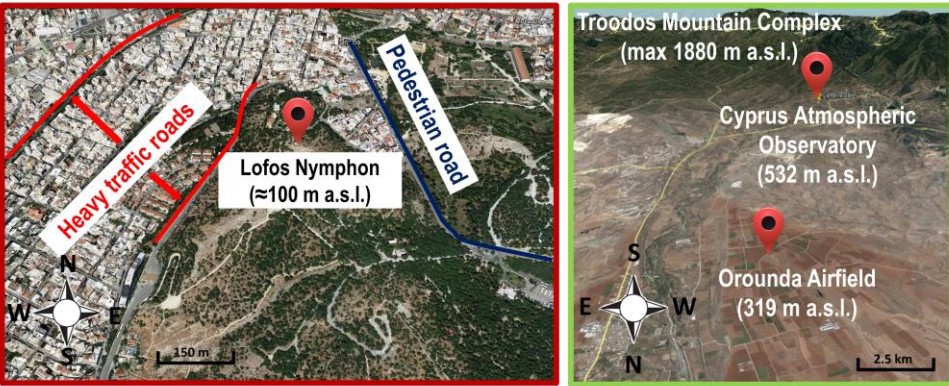

**Figure 2**. Upper panel: Location of the two sampling sites in the Eastern Mediterranean (top). During the Athens campaign, sampling was conducted at Lofos Nymphon (bottom left) surrounded by busy traffic roads (red line) and a touristic area (blue line) free of motor vehicles. During the Cyprus campaign, (bottom right) measurements using UAS was conducted at the Orounda airfield and ground-based monitoring at the Cyprus atmospheric observatory (close to Agia Marina Xyliatou) at the foothills of the Troodos mountains. The elevation difference between these sites is noted. All images are courtesy of Google Maps.

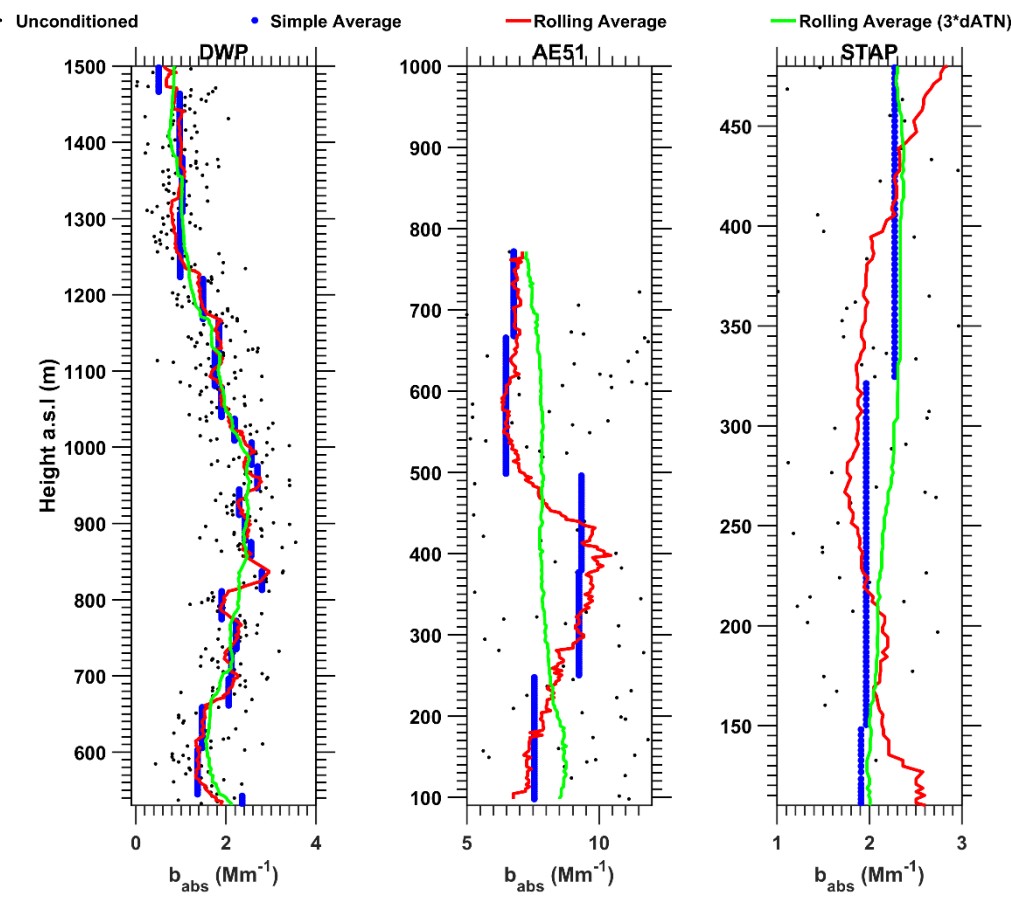

1405

**Figure 3.** Examples of the use of the improved ONA algorithm for the three attenuation monitors examined in this study. Raw data (black dots) are shown against the traditional ONA algorithm (Hagler et al., 2011; blue), the improved ONA using a rolling average and the ΔATN proposed in Section 4 (red), and the improved ONA using the rolling average but with increased ΔATN by a factor of 3 (over-smoothed green). The proposed ΔATN used are 0.01, 0.03, 0.03 for AE51, DWP and STAP, respectively.

1410

1415

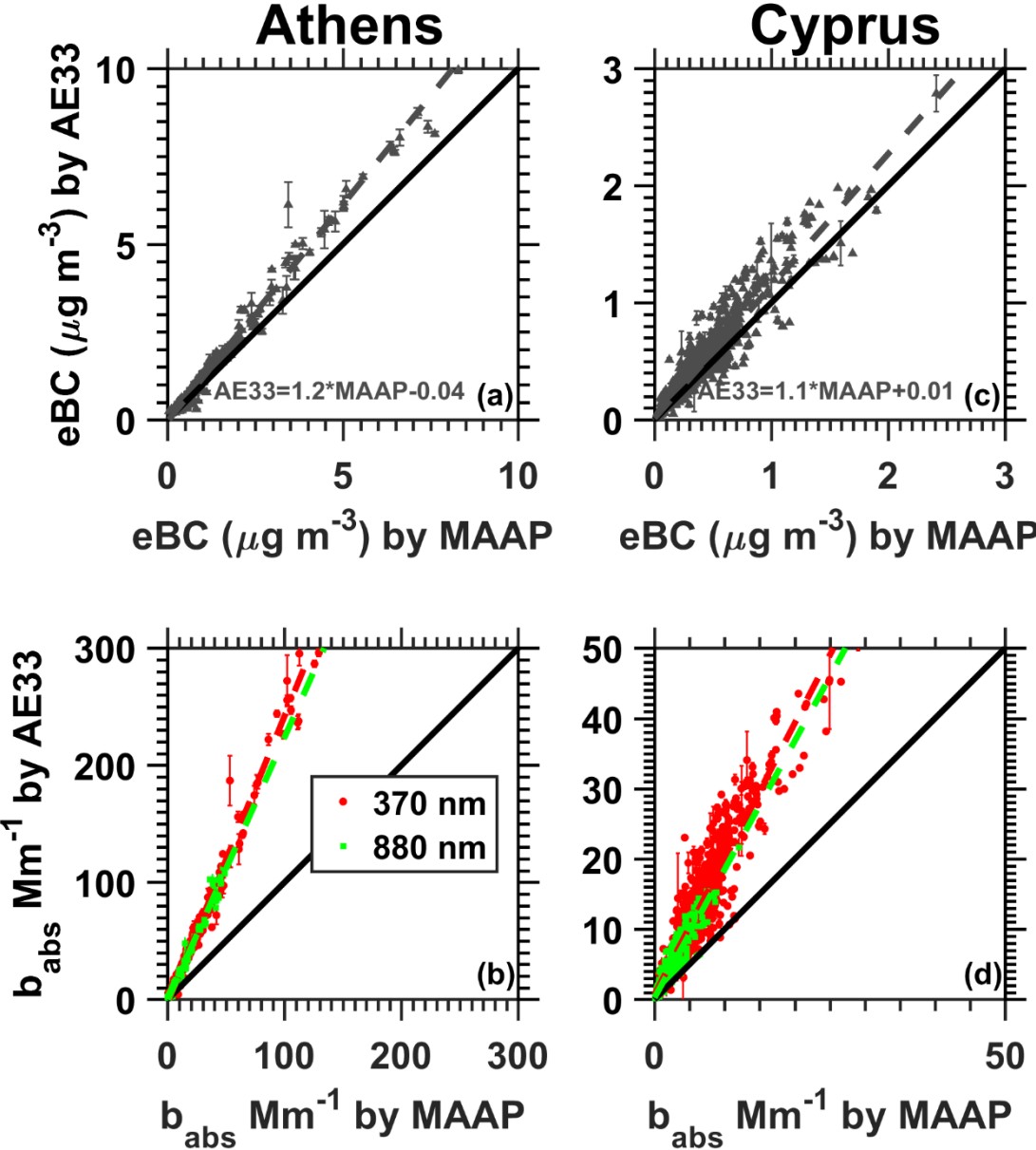

**Figure 4.** Comparison of AE33 against MAAP for eBC (a, c panels) and $b_{abs}$ (b, d panels) at 370nm (red dots) and 880nm (green dots) during the Athens (a, b panels) and Cyprus (c, d panels) campaigns, respectively. Error bars correspond to one standard error from the mean. Not visible error bars suggest that the error estimate is smaller than the area covered by the symbol. The 1:1 and regression lines are shown by a solid black and a dashed line colored accordingly to the instrument, respectively. Results are shown in Table 4.

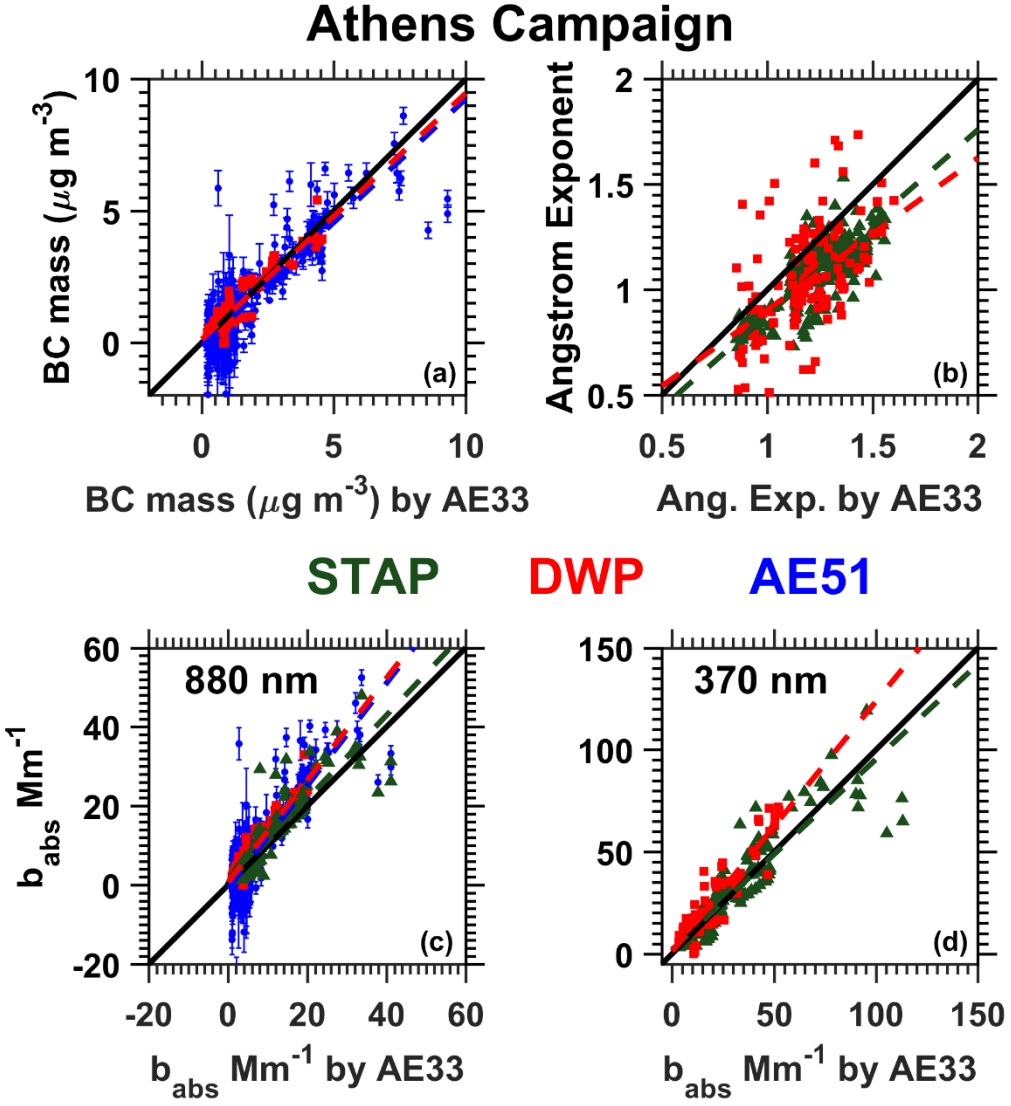

**Figure 5**. Comparison of miniature monitors (STAP: green triangles; DWP: red squares; AE51: blue circles) whilst on the ground against the corrected AE33 during the Athens campaign with respect to *eBC* mass (a), absorption Ångström exponent (b), the absorption coefficient at 880 nm (c) and 370 nm (d). Error bars correspond to one standard error from the mean with respect to AE51. The standard error concerning DWP and STAP with respect $b_{abs}$ and *eBC* is smaller than the symbol in the graph for the vast majority of the cases and is not presented for clarity. The 1:1 and regression lines are shown by a solid black and a dashed line colored accordingly to the instrument, respectively. Results are shown in Table 4.

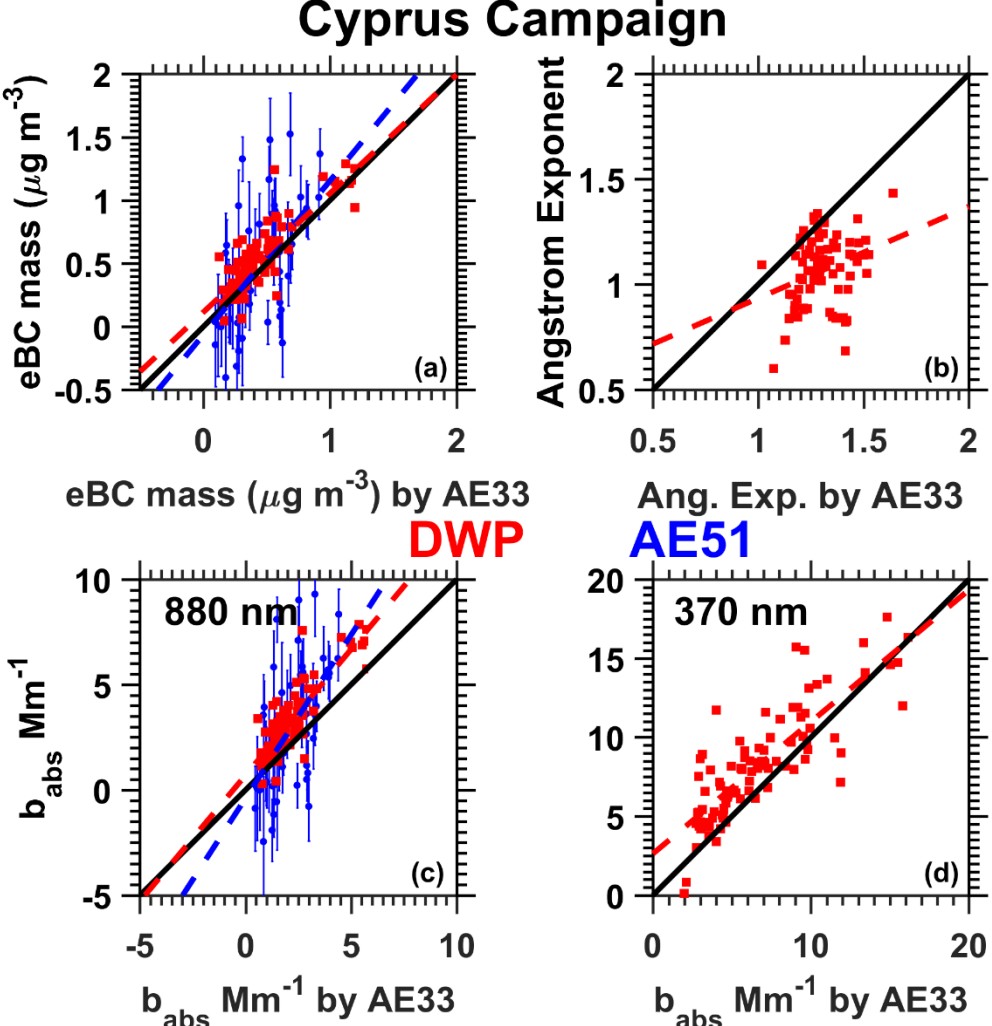

**Figure 6**. Comparison of miniature attenuation monitors (DWP: red squares; AE51: blue circles) while airborne against the corrected AE33 during the Cyprus campaign with respect to eBC mass (a), absorption Ångström exponent (b), and the absorption coefficient at 880 nm (c) and 370 nm (d). Miniature monitors sampled airborne. Error bars correspond to one standard error from the mean with respect to AE51. The standard error concerning DWP and STAP with respect $b_{abs}$ and $eBC$ is smaller than the symbol in the graph for the vast majority of the cases and is not presented for clarity. The 1:1 and regression lines are shown by a solid black and a dashed line colored accordingly to the instrument, respectively. Results are shown in Table 4.

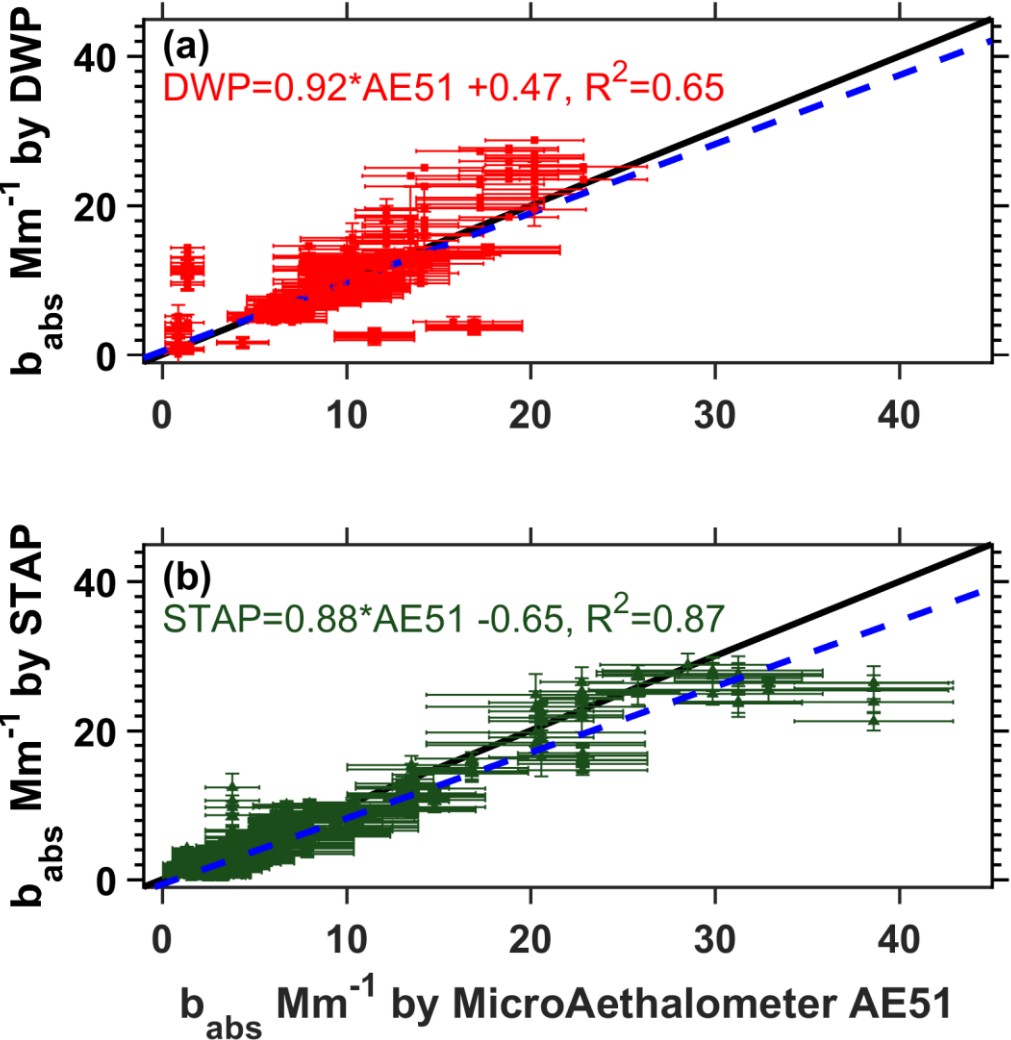

**Figure 7**. Comparison of AE51 against STAP (green triangles) and DWP (red squares) during eight flights of the Athens campaign. The reported agreement in the correlation suggests that no significant bias affected the monitors. The correlation deteriorates ($R^2$=0.01) if data are not processed with the noise reduction algorithm (Section 4). Error bars correspond to one standard error from the mean. Not visible error bars suggest that the smoothing algorithm did not average that sampling point with its neighbors, resulting in a standard deviation and standard error equal to zero. The 1:1 and regression lines are shown by a solid black and a dashed blue line, respectively.

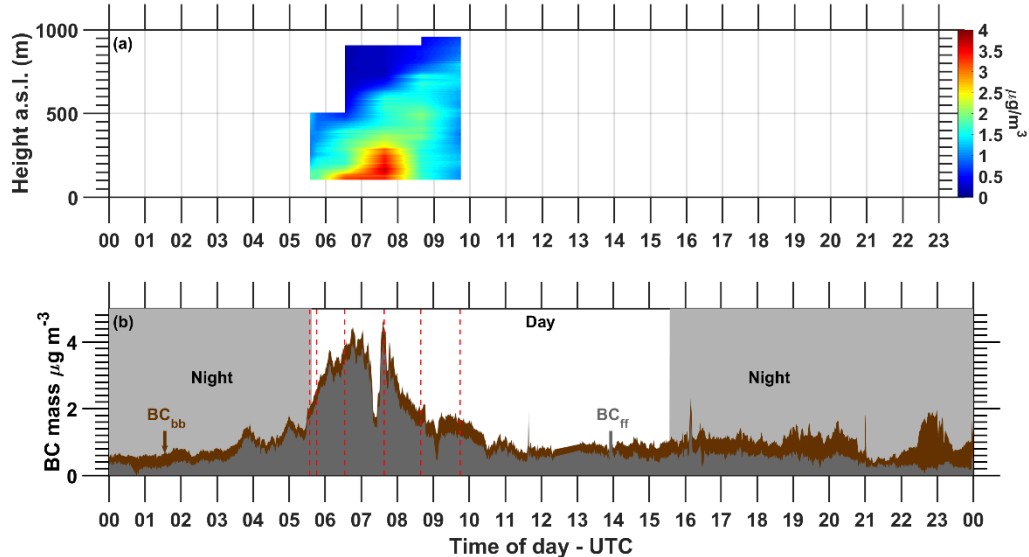

**Figure 8**. Reconstruction of *eBC* mass vertical distribution (a) based on 6 flights between 19th January 2016 (Athens campaign) 05:30 and 09:30 (UTC). The lidar-determined vertical distributions are shown in Fig. 9. The corresponding ground measurements are also shown on panel (b). The concentration of BC from fossil fuel (ff) and biomass burning (bb) are shown with grey and brown color, respectively. Dashed red lines indicate the start of each of the 6 flights the reconstructed *eBC* profiles was based upon.

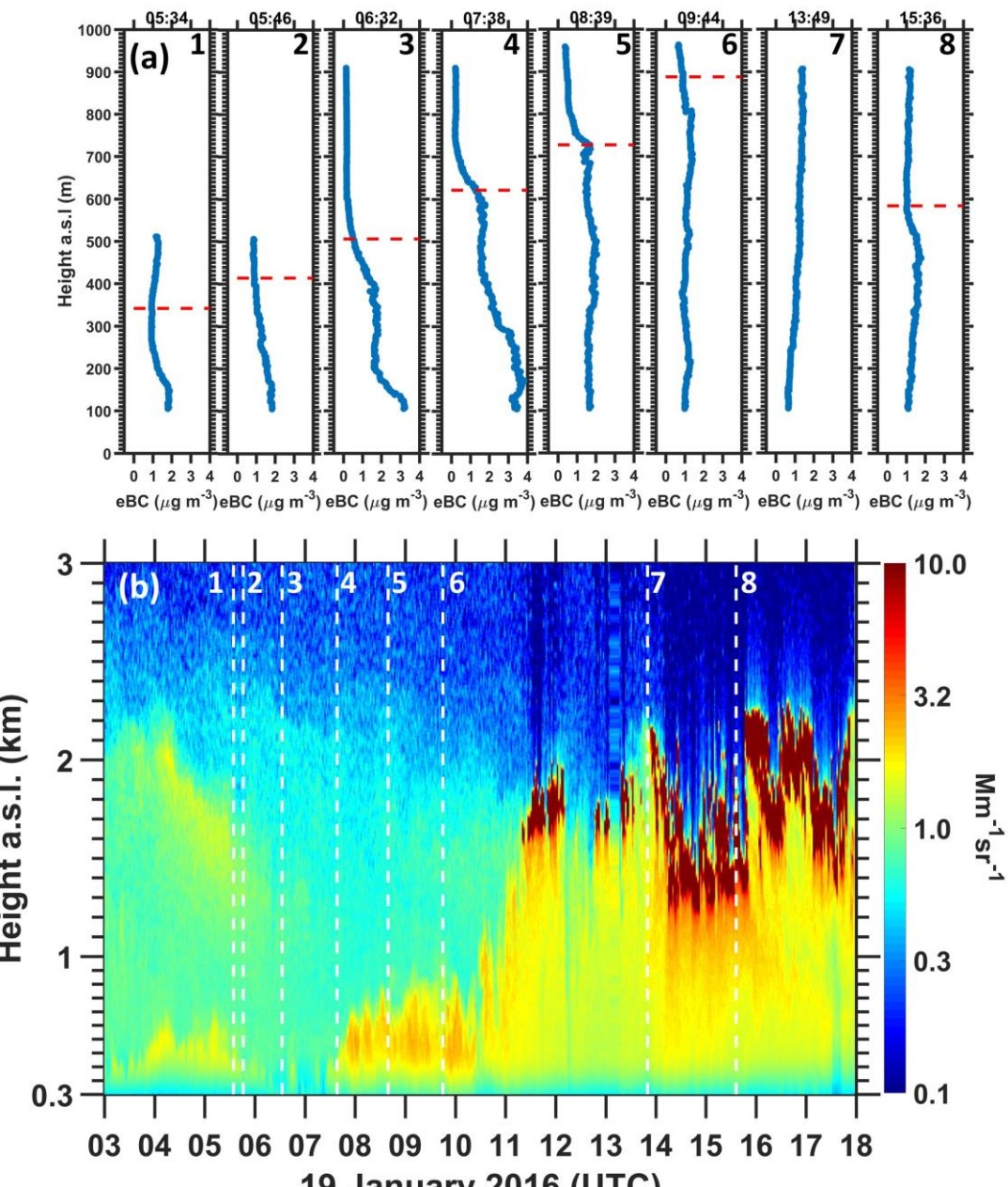

**Figure 9**. Vertical profiles (blue lines) of the *eBC* mass (a), measured during 19[th] January 2016 (Athens campaign) accompanied by the mixing height (dashed red line) of the lower layer derived by Polly-XT measurements. During the 13:49 flight, mixing height was higher than the maximum altitude of flight and it is not shown. The corresponding time-height display of the 1064-nm attenuated backscatter measured with Polly-XT is also shown (b). Dashed white lines correspond to the start of each of the 8 flights performed during that day.