# Peer review of "On-flight intercomparison of three miniature aerosol absorption sensors using Unmanned Aerial Systems (UAS)"

_Atmospheric Measurement Techniques, 2019_

## Referee Comment (RC1) · Anonymous Referee #1 · 22 Jul 2019

This is a very interesting and important manuscript that describes an intercomparison of aerosol light absorption coefficient measurements taken with three miniature instruments for UAS deployment and two "reference" ground-based instruments. While I think that this manuscript describes very high quality and important work, the same cannot be said about the writing style, which could greatly benefit from extensive editing.

Just a few minor comments: this could go on and on:

Lines 85-87: "In fact, the reduced size, weight, and power needs of these systems, along
with the reduced cost of the platforms and instrumentation, make them suit-

able for these
operations with huge potentials currently poorly demonstrated." This statement would benefit from some references.

Lines 87-94: Comments on "endurance and not risking the lives of crew" should be added.

Lines 94-96: "They have also the potential (yet not demonstrated) of the ground-based monitoring networks capabilities in providing long-term atmospheric observations."
This is unclear to me, please reword.

Lines 97-100: How was the "vertical distribution of aerosol absorption" performed???

Lines 347 & 359: "Angstrom" or "angstrom"? How about "Ångström"?

Line 515: "agreement" or should this read "correlation"?

---

## Referee Comment (RC2) · Anonymous Referee #2 · 21 Aug 2019

The manuscript fits within the scope of Atmospheric Measurement Techniques and the insights are novel enough to justify publication. There are though a number of details that should be addressed before the publication and the manuscript should be thoroughly checked.

The authors use an atypical structure (sections) for AMT, or any journal for that matter. This is fine but it would be critical to detail in the text when something gets explained in a later section. e.g. The field campaigns should somehow mention that the UAS will discussed later as one expect to read details when something like multicopter comes up. Alternatively may be the authors should consider to first discuss the tools, then the

platforms before discussing the field campaigns as the primary focus here should be on tools, platforms and measurements not on the field campaigns.

The current state of the art and background are extremely poorly described in the introduction. The paper is very misleading on how novel BC measurements by UAV are. It gives an impression that this is very novel when in reality BC has been measured on UAVs for more than 10 years (see Corrigan et al., 2008). This is just not proper. You mention Bates for the STAB but fail to mention that the Bates paper is not only this instrument but this instrument on a UAV. Your paper has to acknowledge what is out there, what instruments flew on what platforms etc.. Also in regards to the discussion of the vertical profile observations, there are studies to compare to, both UAV and balloon (besides Bates, Corrigan, there Po Valley Ferrero et al., 2014).

Some statements are misleading and/or too qualitative in the discussion. The authors confuse correlation with agreement (Line 513 discussion figure 4), MAAP and AE33 are not in agreement if there is a 20% bias. The measurements are well correlated but the values are substantially off, systematically yes but still the values are not in agreement at all. Also, the authors use too many qualitative statements like "excellent" when it is unclear what excellent means. Things ae statistically significant or not.

Finally, one has to hope that the authors were more careful in their experiments than in the preparation of the manuscript. The manuscript needs a serious re-read with attention to detail for text formatting, typos, format of references and completeness of references cited. A few items are in details but the list is not certainly exhaustive.

Details

L29-31 "the measured signal of the three sensors was converted into absorption co-efficient, …. and, when applicable, to signal saturation corrections following the suggestions of the manufacturers." Please reformulate, the signal was not converted to corrections but you applied corrections

L58 redefine abbreviations at first use in the main text (here BC)

L64-69 This is a poor representation of the existing methods and partly misleading... The sunset is thermal AND optical and there are thermal optical transmission and thermal optical reflection... see AMT papers on the subjects. For just evolved gas phase there are the old commercial systems such as the R&P analyzer, there is the DRI analyzer. You only give 2 Sunset papers. This is not critical but just weird and actually wrong.

L68 Please check also all your references throughout the manuscript. Here the Petzold and Moosmueller refs are both missing in the literature cited.

L123 Athens campaign. If you keep the structure with first field campaigns then instruments then please reference at the mention of multicopter that you will provide details later, idem later on for the Cyprus study UAS.

L128 The 2kg payload limitation is confusing as the Table says different. Please elaborate and please elaborate and clarify which instrument this refers to.

L371 section on miniature monitor descriptions. Please discuss them all 3 Currently hardly any description is here on the DWP and please be consistent by providing/discussing weight of all 3 of them. Essentially give the same information and same level of detail for all 3 consistently. This would be most useful.

The UAS platforms in Table 1 should include manufacturer.

Table 1: typo km not k, formatting: align text to center of pictures

Table 1. define abbreviations at first use

Figure 1: provide the source of the maps and pictures and ensure you have the rights

Figures 4,5,6,7: could you provide error bars on the values. If they are smaller than the symbols used then please state so in the legend.

Figure 7 Why do the x and y axis extend to negative values?

Literature cited:

Please check all references are included

Please format uniformly, especially year, some years are missing sometimes year is after the author, sometimes year is at the end

Curious that besides the author's own papers there is no citation more recent than 2016. There seem to be relevant literature out there.. e.g. Saturno et al., 2017 on aethalometer correction schemes

References mentioned in this review

Bates, T. S., Quinn, P. K., Johnson, J. E., Corless, A., Brechtel, F. J., Stalin, S. E., Meinig, C., and Burkhart, J. F.: Measurements of atmospheric aerosol vertical distributions above Svalbard, Norway, using unmanned aerial systems (UAS), Atmos. Meas. Tech., 6, 2115-2120, https://doi.org/10.5194/amt-6-2115-2013, 2013.

Corrigan, C. E., Roberts, G. C., Ramana, M. V., Kim, D., and Ramanathan, V.: Capturing vertical profiles of aerosols and black carbon over the Indian Ocean using autonomous unmanned aerial vehicles, Atmos. Chem. Phys., 8, 737-747, https://doi.org/10.5194/acp-8-737-2008, 2008.

Ferrero, L., Castelli, M., Ferrini, B. S., Moscatelli, M., Perrone, M. G., Sangiorgi, G., D'Angelo, L., Rovelli, G., Moroni, B., Scardazza, F., Močnik, G., Bolzacchini, E., Petitta, M., and Cappelletti, D.: Impact of black carbon aerosol over Italian basin valleys: high-resolution measurements along vertical profiles, radiative forcing and heating rate, Atmos. Chem. Phys., 14, 9641-9664, https://doi.org/10.5194/acp-14-9641-2014, 2014.

Saturno, J., Pöhlker, C., Massabò, D., Brito, J., Carbone, S., Cheng, Y., Chi, X., Ditas, F., Hrabě de Angelis, I., Morán-Zuloaga, D., Pöhlker, M. L., Rizzo, L. V., Walter, D., Wang, Q., Artaxo, P., Prati, P., and Andreae, M. O.: Comparison of different Aethalome-

ter correction schemes and a reference multi-wavelength absorption technique for ambient aerosol data, Atmos. Meas. Tech., 10, 2837-2850, https://doi.org/10.5194/amt-10-2837-2017, 2017.

---

## Author Comment (AC1) · 1 Oct 2019

The authors would like to thank the reviewers for their constructive comments. Please find our responses below.

Point 1: This is a very interesting and important manuscript that describes an intercomparison of aerosol light absorption coefficient measurements taken with three miniature instruments for UAS deployment and two "reference" ground-based instruments. While I think that this manuscript describes very high quality and important work, the same cannot be said about the writing style, which could greatly benefit from extensive editing.

[Figure]

Following the advice of the reviewer, the manuscript has been re-edited and several typos were corrected. The structure of the manuscript has been altered as suggested with the "Instrumentation" section moved forward for clarity. Several sentences that were deemed confusing were omitted or changed, especially those speculating on the potential of UAS capabilities.

Point 2: Lines 85-87: "In fact, the reduced size, weight, and power needs of these systems, along with the reduced cost of the platforms and instrumentation, make them suitable for these operations with huge potentials currently poorly demonstrated." This statement would benefit from some references.

This sentence was confusing and has been removed. Additionally, the referenced literature related to UAS-based absorption has been expanded.

Point 3: Lines 87-94: Comments on "endurance and not risking the lives of crew" should be added.

This sentence has been removed from the revised manuscript.

Point 4: Lines 94-96: "They have also the potential (yet not demonstrated) of the ground-based monitoring networks capabilities in providing long-term atmospheric observations." This is unclear to me, please reword.

Once more, this sentence seems to have caused confusion and has been omitted.

Point 5: Lines 97-100: How was the "vertical distribution of aerosol absorption" performed???

This is now clarified in the revised manuscript. This sentence now reads "we focus on vertical distributions of aerosol absorption, measured with miniature absorption sensors onboard small and medium-size UAS during two intensive field campaigns at contrasting locations in the Eastern Mediterranean". This can be found in lines 112-116 of the revised manuscript.

Point 6: Lines 347 & 359: "Angstrom" or "angstrom"? How about "Ångström"?

This word has been replaced, as suggested by the reviewer, in the revised manuscript

Point 7: Line 515: "agreement" or should this read "correlation"?

In the first version of the manuscript, the term "agreement" was confusing and has been replaced by the term "correlation" throughout as suggested.
* * *

---

## Author Response (AR1)

The authors would like to thank the reviewers for their constructive comments. Please find our responses below.

**Reviewer #1**

This is a very interesting and important manuscript that describes an intercomparison of aerosol light absorption coefficient measurements taken with three miniature instruments for UAS deployment and two "reference" ground-based instruments. While I think that this manuscript describes very high quality and important work, the same cannot be said about the writing style, which could greatly benefit from extensive editing.

Following the advice of the reviewer, the manuscript has been re-edited and several typos were corrected. The structure of the manuscript has been altered as suggested with the "Instrumentation" section moved forward for clarity. Several sentences that were deemed confusing were omitted or changed, especially those speculating on the potential of UAS capabilities.

Lines 85-87: "In fact, the reduced size, weight, and power needs of these systems, along with the reduced cost of the platforms and instrumentation, make them suitable for these operations with huge potentials currently poorly demonstrated." This statement would benefit from some references.

This sentence was confusing and has been removed. Additionally, the referenced literature related to UAS-based absorption has been expanded.

Lines 87-94: Comments on "endurance and not risking the lives of crew" should be added.

This sentence has been removed from the revised manuscript.

Lines 94-96: "They have also the potential (yet not demonstrated) of the ground-based monitoring networks capabilities in providing long-term atmospheric observations." This is unclear to me, please reword.

Once more, this sentence seems to have caused confusion and has been omitted.

Lines 97-100: How was the "vertical distribution of aerosol absorption" performed???

This is now clarified in the revised manuscript. This sentence now reads "we focus on vertical distributions of aerosol absorption, measured with miniature absorption sensors onboard small and medium-size UAS during two intensive field campaigns at contrasting locations in the Eastern Mediterranean". This can be found in lines 112-116 of the revised manuscript.

Lines 347 & 359: "Angstrom" or "angstrom"? How about "Ångström"?

This word has been replaced, as suggested by the reviewer, in the revised manuscript

Line 515: "agreement" or should this read "correlation"?

In the first version of the manuscript, the term "agreement" was confusing and has been replaced by the term "correlation" throughout as suggested.

**Reviewer #2**

The manuscript fits within the scope of Atmospheric Measurement Techniques and the insights are novel enough to justify publication. There are though a number of details that should be addressed before the publication and the manuscript should be thoroughly checked.

The authors use an atypical structure (sections) for AMT, or any journal for that matter. This is fine but it would be critical to detail in the text when something gets explained in a later section. e.g. The field campaigns should somehow mention that the UAS will discussed later as one expect to read details when something like multi-copter comes up. Alternatively may be the authors should consider to first discuss the tools, then the platforms before discussing the field campaigns as the primary focus here should be on tools, platforms and measurements not on the field campaigns.

The authors agree that the form of the manuscript did not assist the reader. In the revised manuscript the "Instrumentation" section now precedes that of the "Campaigns".

The current state of the art and background are extremely poorly described in the introduction. The paper is very misleading on how novel BC measurements by UAV are. It gives an impression that this is very novel when in reality BC has been measured on UAVs for more than 10 years (see Corrigan et al., 2008). This is just not proper. You mention Bates for the STAB but fail to mention that the Bates paper is not only this instrument but this instrument on a UAV. Your paper has to acknowledge what is out there, what instruments flew on what platforms etc.. Also in regards to the discussion of the vertical profile observations, there are studies to compare to, both UAV and balloon (besides Bates, Corrigan, there Po Valley Ferrero et al., 2014).

The above statement is a misunderstanding. The authors consider novel the use of miniature sensors onboard small size UAS. Small UAS are considered those with a gross weight smaller than 25 kg as suggested by the FAA. In the revised manuscript, there is a clearer description of this notion so to avoid confusion. The term "small UAS" is used and in Section 2.1 this is clarified. Furthermore, there is a distinction between standard (rack) size and small sensors (Lines 82-88 of the revised manuscript).

Taking into account the above comment, we have substantially edited the relevant paragraph in the introduction (see revised manuscript) and included several publications concerning absorption vertical profiles, either using UAS or tethered balloons. (Lines 82-96).

It should be noted that the main focus of this work is to present the instrument intercomparison. A comparison between the findings from the campaigns described in this work and those already found in the literature is beyond the scope of this study. Our intention is to focus on the results of both campaigns in a future separate publication.

Some statements are misleading and/or too qualitative in the discussion. The authors confuse correlation with agreement (Line 513 discussion figure 4), MAAP and AE33 are not in agreement if there is a 20% bias. The measurements are well correlated but the values are substantially off, systematically yes but still the values are not in agreement at all. Also, the authors use too many qualitative statements like "excellent" when it is unclear what excellent means. Things are statistically significant or not.

The reviewer is correct. The term "agreement" has been replaced by the term "correlation" throughout the manuscript. Additionally, a Welch's t-test, used in cases when populations of unequal variances are to be compared, was employed to deduct the significance of the differences discussed in this work.

Finally, one has to hope that the authors were more careful in their experiments than in the preparation of the manuscript. The manuscript needs a serious re-read with attention to detail for text formatting, typos, format of references and completeness of references cited. A few items are in details but the list is not certainly exhaustive.

Hewewith, we would like to mention that the high-quality of the data presented in our work has been assured in several ways (reference instruments, sample conditioning (dryer), airborne and inflight intercomparison). The authors acknowledge indeed that several small errors were present in the old manuscript, which has undergone extensive re-editing. This includes, among others; correction of typos, confusing sentences to be rephrased or omitted, uniform spacing and text formatting throughout.

L29-31 "the measured signal of the three sensors was converted into absorption coefficient,... and, when applicable, to signal saturation corrections following the suggestions of the manufacturers." Please reformulate, the signal was not converted to corrections but you applied corrections.

This sentence has been edited and in the revised manuscript and now reads at Line 31:" The measured signal of the miniaturized sensors was converted into the absorption coefficient and equivalent black carbon concentration ($eBC$). When applicable, signal saturation corrections were applied, following the suggestions of the manufacturers."

L58 redefine abbreviations at first use in the main text (here BC)

BC is now defined in Line 56. The manuscript was checked to ensure that each abbreviation was defined at first use. The authors include a nomenclature table to assist the reader with the abbreviations employed in the text.

L64-69 This is a poor representation of the existing methods and partly misleading ... The sunset is thermal AND optical and there are thermal optical transmission and thermal optical reflection... see AMT papers on the subjects. For just evolved gas phase there are the old commercial systems such as the R&P analyzer, there is the DRI analyzer. You only give 2 Sunset papers. This is not critical but just weird and actually wrong.

This sentence was reformulated by removing the reference of the two Sunset papers, mentioning briefly the thermal-optical techniques, which are now replaced bya reference to Lack et al., 2014, a review paper that describes in-depth the techniques for measuring the concentration of black carbon.

L68 Please check also all your references throughout the manuscript. Here the Petzold and Moosmueller refs are both missing in the literature cited.

We thank the reviewer for identifying this omission. The revised manuscript was thoroughly checked. All references were additionally checked for abiding with the AMT format.

L123 Athens campaign. If you keep the structure with first field campaigns then instruments then please reference at the mention of multicopter that you will provide details later, idem later on for the Cyprus study UAS.

This structure has changed in the revised manuscript, with the instrumentation section preceding that of the campaign.

L128 The 2kg payload limitation is confusing as the Table says different. Please elaborate and please elaborate and clarify which instrument this refers to.

The reviewer is correct as these two statements correspond to different payloads. Line 407 in the revised manuscript (used to be Line 128) refers to the instrument only assuming an extra 2 kg for batteries, the dryer, and the inlet. In Table 1 the net payload was specified. The authors acknowledge that this sentence was confusing and have rephrased it to "Due to payload restrictions (2 kg maximum for scientific instrumentation and another 2 kg payload for the batteries, dryer and inlet),…".

L371 section on miniature monitor descriptions. Please discuss them all 3 Currently hardly any description is here on the DWP and please be consistent by providing/discussing weight of all 3 of them. Essentially give the same information and same level of detail for all 3 consistently. This would be most useful.

As proposed, information on DWP has been added. In specific, the following sentences are now found in Lines 329-333 of the revised manuscript "The DWP has been constructed as a modification of the AE51, by placing an additional light source, emitting at 370 nm. Additionally, the sampling flow rate has been increased to 2 L min$^{-1}$, by replacing the original AE51 pump, with an external whose flow rate is controlled by a critical orifice. The external pump resulted in additional weight to DWP. "

The UAS platforms in Table 1 should include manufacturer.

This information has been included in the revised manuscript

Table 1: typo km not k, formatting: align text to center of pictures

The typo has been corrected

Table 1. define abbreviations at first use

These have been defined and have been included in the nomenclature table.

Figure 1: provide the source of the maps and pictures and ensure you have the rights

Figure 1 (formerly Fig. 2) is owned by the authors. Figure 2 (formerly Fig. 1) is owned by Google which encourages the use of its maps for scientific publication. Google policy can be summarized as follows "Due to limited resources and high demand, we're

unable to sign any letter or contract specifying that your project or use has our explicit permission. As long as you follow the guidance on this page, and attribute the content correctly, feel free to move forward with your project." Which can be found at https://www.google.com/permissions/geoguidelines/

Figures 4,5,6,7: could you provide error bars on the values. If they are smaller than the symbols used then please state so in the legend.

Error bars have been added in Fig. 4,5,6,7 corresponding to one standard error from the mean. It is noted that if one standard deviation from the mean was plotted then Fig 5, 6, and 7 would become unreadable; mainly because AE51 is quite noisy. Therefore, to keep a uniform and concise format throughout, all figures exhibit one standard error from the average, while standard deviation is reported in the text. Under this configuration, the symbols concerning STAP and DWP in Fig. 5 and 6 are larger than the error bars, which is denoted in the legend of each figure.

**List of changes applied in the revised manuscript**

The original manuscript has undergone extensive editing, as can be seen in the tracked changes version found below.

Major changes include:

- **Text**

Several references were added in the revised manuscript concerning vertical absorption profiles using UAS or balloons (Line 90-96), manned aircrafts (Line 79-81), mathematical tests applied to examine the statistical significance of the reported differences (Line 545=550) and a recent publication on the discrepancies of RH on absorption measurements (Line 538).

The section "Instrumentation" precedes that of "Sampling sites".

Abbreviated physical properties such as *eBC*, $b_{abs}$ are now all in Italic.

Line 448-455. A paragraph referring to LIDAR measurements has been moved from the "Ground based reference instruments" to the "Cyprus Campaign"

Line 329-339: The discussion regarding DWP has been expanded.

Lines 563-569: An explanation is provided as to why small differences in eBC measurements from AE33 and MAAP grow large with respect to the absorption coefficient. In addition the revised manuscript the values used to derive the absorption coefficient are now mentioned (see eg Line 700-701)

Lines 839-845: One of the basic findings of this work is now further analyzed in the "Conclusions"

- **Figures**

Figure 1 in the original manuscript is listed in the revise manuscript as Fig. 2

Figure 2 in the original manuscript is listed in the revise manuscript as Fig. 1

Figure 2 denotes source of the images

Figures 4, 5, 6 and 7 include error bars. The caption of each figure is also updated

Fig. 4 includes information on the slope of the comparison

Fig 9 lower panel now shows attenuated backscatter instead of arbitrary signal

A new figure showing the comparison of MAAP against AE33 at 637 nm is included in the supplement. It has been listed as Suppl. Fig. 2. In the revised manuscript Suppl. Fig 2 is now listed as Suppl. Fig. 3, Suppl. Fig 3 is now listed as Suppl. Fig. 4, etc.

- **Tables**

Table 1 includes information on the manufacturer as suggested by Reviewer #2.

Table 1 caption includes information on UAS size.

MTOW is now defined as a footnote of Table 1.

In the revised manuscript, Table 2 first column is "Instrument name" and the second "Manufacturer"

[revised manuscript text omitted]